# Monsoonal influence on floating marine litter pathways in the Bay of Bengal

Lianne C. Harrison[1,2], Jennifer A. Graham[1,2], Piyali Chowdhury[1,2], Tiago A. M. Silva[1,2], Danja P. Hoehn[1,2], Alakes Samanta[3], Kunal Chakraborty[3], Sudheer Joseph[3], T. M. Balakrishnan Nair[3], T. Srinivasa Kumar[3]

[1]Centre for Environment, Fisheries and Aquaculture Science, Lowestoft, NR33 0HT, UK
[2]Collaborative Centre for Sustainable Use of the Seas, University of East Anglia, Norwich, NR4 7TJ, UK
[3]Indian National Centre for Ocean Information Services, Ministry of Earth Sciences, Hyderabad, India

*Correspondence to*: Lianne C. Harrison (lianne.harrison@cefas.gov.uk)

**Abstract.** Marine litter in the Bay of Bengal has been under-studied despite large quantities of mismanaged waste reportedly entering the ocean from its surrounding countries. The seasonal reversal of monsoon currents in this region provides a unique environment for the transport of floating macro-litter. A particle tracking model is used here to investigate source-to-sink connectivity of marine debris between countries via oceanic pathways in the Bay of Bengal. We use an approach considering uniform release of particles along the entire coastline, avoiding the considerable uncertainties associated with assumed riverine sources. Two different simulations are considered, forced with either a high-resolution ocean hindcast developed specifically for the Bay of Bengal or a lower-resolution dataset which includes data assimilation. The vast majority of particles released during our simulations were found to beach within 16 months; most particles beached in their country of origin (57-90%), with connectivity towards Myanmar accounting for the second highest connectivity rates (2-29%) from many countries within the Bay of Bengal. This is likely due to the relatively large size of Myanmar's coastline and that it lies in the path of the East India Coastal Current for much of the year (February-September). Patterns of connectivity were found to change along with the monsoon, and the post-monsoon period (October-January) showed a notably greater dispersal of particles than the rest of the year. Both simulations were evaluated using the pathways of undrogued surface drifters, which moved primarily within the open ocean, with better agreement found here for particles advected by data-assimilated ocean velocities. This study will therefore crucially inform future research and policy in this region, providing advice on the benefits and suitability of selecting different modelling approaches independent of assumptions of the source locations or volumes.

## 1 Introduction

Marine litter is a worldwide concern that is being widely investigated in an effort to mitigate ecosystem effects, such as entanglement and ingestion by marine animals and physical damage to delicate habitats like coral reefs (Gall and Thompson, 2015). Plastic pollutants have formed the focus of these investigations due to their abundance and longevity within the marine environment. Jambeck et al. (2015) estimated that between 4.8 - 12.7 million tonnes of plastic entered our oceans every year

(based on conditions in 2010), and that this could increase by an order of magnitude by 2025. Other studies have calculated significantly different estimates for how much plastic finds its way to the ocean. Lebreton and Andrady (2019) calculated a lower estimate of between 3.1 – 8.2 Mt of plastic entering the ocean each year, using a different dataset for solid waste

generation but similar assumptions to that of Jambeck et al. (2015) about how much mismanaged waste within 50 km of the coast finds its way into the ocean. Several other studies have investigated slightly different questions and about how much plastic waste enters the oceans which have resulted in quite different estimates. Lebreton et al. (2017), Schmidt et al. (2018) and Meijer et al. (2021) calculated how much plastic is transported solely by rivers to the ocean, resulting in much lower values of 1.1 - 2.4 Mt/yr, 0.5 – 2.8 Mt/yr, and 0.8 – 2.7 Mt/yr, respectively. A subsequent study by Borrelle et al. (2020) found a

significantly larger value of 19 – 23 Mt of plastic waste ending up in aquatic environments in 2016, however, this includes rivers and lakes rather than just the ocean. Despite the large uncertainties associated with these estimates, observations of so-called 'garbage patches' that have formed in the ocean's major gyres (Cózar et al., 2014; Eriksen et al., 2014) and reports of litter washing up on beaches (e.g. Shankar et al., 2023) confirm plastic pollution in the ocean is a vast problem.

      Modelling the transport of marine debris has been used to determine the sources and sinks of pollutants and inform

policies aiming to reduce the accumulation of marine debris in the ocean and along coastlines. Much of the previous marine litter transport modelling has been done on a global scale (Chassignet et al., 2021; Chenillat et al., 2021; Eriksen et al., 2014; Isobe and Iwasaki, 2022; Lebreton et al., 2012) and many concentrate on how much litter remains within the ocean garbage patches. However, multiple recent studies have suggested that approximately two-thirds to three-quarters of all litter released in global model simulations may be captured on coastlines, though they note there are large uncertainties associated with these

estimates (Chassignet et al., 2021; Chenillat et al., 2021; Lebreton et al., 2019; Onink et al., 2021).

      The Bay of Bengal was found by Chassignet et al. (2021) and Lebreton et al. (2012) to have among the highest concentrations of floating litter in their global simulations, yet only a couple of modelling studies published to date have dealt specifically with this region (Irfan et al., 2024; van der Mheen et al., 2020a). Irfan et al. (2024) examined the effects of windage and Stokes drift velocities on the locations and propensity of particles to 'beach', or wash ashore, at different times of the year.

'Windage' refers to the direct influence of wind velocities on the portions of buoyant litter found above the surface of the ocean; the larger or more buoyant the items are, the greater its effect. Stokes drift accounts for the net movement of particles due to the motion of waves. Irfan et al. (2024) highlighted that both mechanisms were crucial to trapping particles in the northern Indian Ocean. They concluded that beaching in the Bay of Bengal peaked on the north-northeast coastlines during the Southwest Monsoon (which they defined as June - October) but did not quantify beaching rates for each country. van der

Mheen et al. (2020a) simulated floating plastic debris in the northern Indian Ocean, identifying monsoonal transport between the Arabian Sea and the Bay of Bengal. They found coastlines in the Bay of Bengal in particular suffered high rates of beaching in their simulations because of the large amounts of plastic waste originating from countries in the region combined with ocean currents pushing buoyant plastic debris into the Bay of Bengal.

      While van der Mheen et al. (2020a) did analyse the connectivity of litter pathways between countries, their model did

not include windage or Stokes drift which have been shown to be important for beaching behaviour of buoyant litter (Irfan et

al., 2024). Additionally, both studies seeded particles in their simulations from river locations based on estimates of waste input into the ocean by Lebreton et al. (2017), which have very high uncertainties (Chassignet et al., 2021; Meijer et al., 2021). Therefore, there is a knowledge gap concerning estimates of litter transfer from wider sources between countries within the Bay of Bengal. Although it is a global problem, understanding the processes connecting sources of marine litter to their sinks in this region is important given the significant evidence suggesting that large amounts of litter are released from Asian countries, in part due to dense populations and lack of waste management infrastructure (Chenillat et al., 2021; Jambeck et al., 2015; Lebreton et al., 2017; Meijer et al., 2021).

The Bay of Bengal is expected to have significantly different trends in litter pathways at different times of the year, due to the seasonal reversal of winds and associated ocean currents. The dominant surface currents in the Bay of Bengal are the Northeast and Southwest Monsoon Currents, and East India Coastal Current (EICC) (Fig. 1). These all vary seasonally, except for the southern branch of the EICC which flows past the east coast of Sri Lanka and remains southward throughout the year. The northern section of the EICC (north of ~10°N) travels south-westwards, along with the southern component, between November – January, before changing direction and flowing north-eastwards for the rest of the year. This current is strongest during spring and transforms into a series of eddies that line the eastern Indian coastline during the summer. The Northeast Monsoon Current flows westward, past the southern tip of India and Sri Lanka, during winter. This current reverses and becomes the Southwest Monsoon Current in summer, flowing eastward in the same location. The Sri Lanka Dome appears during summertime when the Southwest Monsoon Current passes the southern coast of Sri Lanka and swings around the southward-flowing southern component of the EICC to join up with the northward-flowing northern component at approximately 10°N (Fig. 1a). The reader is referred to a review by Phillips et al. (2021) for a full description of the currents and their drivers in this region.

This study used a Lagrangian particle tracking model to investigate the connectivity between six countries: Sri Lanka, India, Bangladesh, Myanmar, Thailand, & Indonesia (Fig. 1), with a focus on understanding source-to-sink dynamics of floating macro-litter within the Bay of Bengal, independent of the size of the sources of litter.

## 2 Methods

The transport of marine litter was modelled here using the OceanParcels v2.3.1 Lagrangian particle tracking model (Delandmeter and van Sebille, 2019; Lange and van Sebille, 2017). The model includes several processes which are believed to be the main physical processes responsible for influencing the movement of floating particles around the domain to simulate the dispersal of buoyant marine debris (Haza et al., 2019). This approach follows similar methods of others to simulate marine plastics distribution (e.g. Chassignet et al., 2021; Isobe and Iwasaki, 2022). Advection of particles via surface ocean currents (detailed below) was included using an inbuilt OceanParcels kernel which uses a fourth-order Runge-Kutta advection scheme (Advection RK4, described in Lange and van Sebille, (2017)). Stokes drift velocities were included to account for the movement of particles resulting from wave motions by simple addition to surface current velocities. To account for sub-grid scale processes, diffusion is implemented as a random walk, through an inbuilt kernel known as DiffusionUniformKh. A

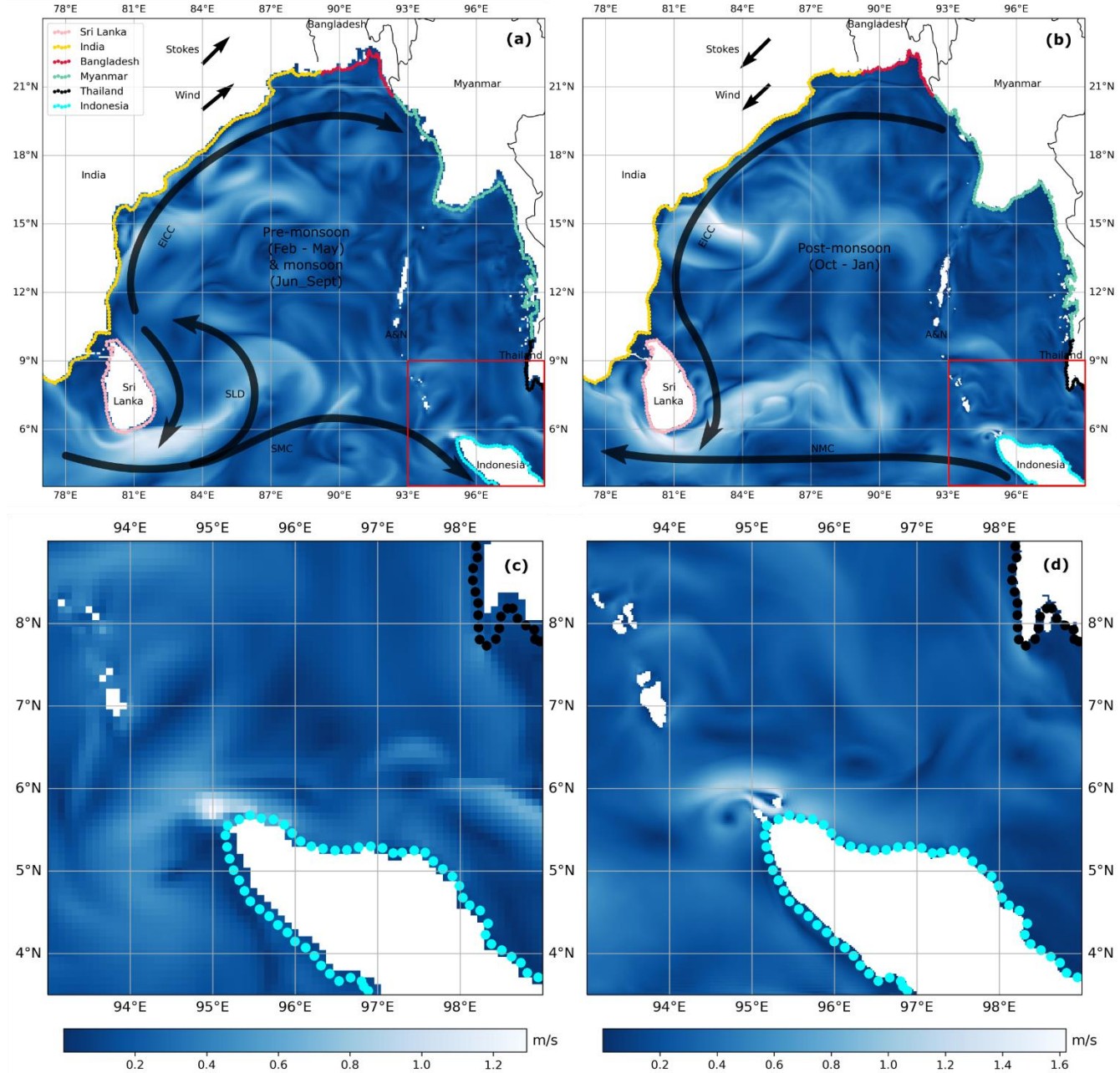

**Figure 1: Major currents in the Bay of Bengal during the pre-monsoon (Feb - May) and monsoon seasons (Jun - Sept) (a) and the post-monsoon season (Oct - Jan) (b). EICC is East India Coastal Current; SLD is Sri Lanka Dome; SMC is Southwest Monsoon Current; NMC is Northeast Monsoon Current; A&N is Andaman and Nicobar Islands. Arrows of Stokes drift and wind velocities show the mean direction averaged over each time period; magnitudes are arbitrary. Blue backgrounds show ocean current speeds on 24th July 2018 from the CMEMS model (a) and ROMS model (b) and areas in the red box are enlarged in (c) and (d), respectively. Coloured markers indicate particle release locations and are colour-coded by country of origin.**

diffusion coefficient of 100 m²/s was chosen based on grid cell size (Peliz et al., 2007), as detailed below. The diffusion kernel combined this coefficient with a random variate calculated from a normal distribution with a mean of zero and standard deviation equal to the square root of the model timestep (see https://doi.org/10.5281/zenodo.14906471 where a copy of the advection and diffusion kernels used have been archived). Windage is implemented in the model by applying 1% of the wind velocity to the particles' trajectories. Following analysis of observations of the wind's effect on undrogued drifters by Pereiro et al. (2018), this should describe all except very buoyant items of litter. While the effect of wind on the surface ocean currents is already included in the ocean velocities used in the particle tracking simulations, the addition of windage takes into account the extra push that wind provides as a result of friction against the portion of marine debris that extends above the surface. The final process implemented here was beaching. At the end of each timestep, after advancing each particle's position, ocean velocities were checked at this new position. If the ocean velocity was less than $10^{-14}$ m/s, the particle was considered to be beached (after Delandmeter and van Sebille (2019)) and was no longer tracked. Stokes drift and wind velocities were not included in the calculations to determine if a particle was beached. Figure A1 shows the final locations of beached particles with respect to the land in each of the model grid resolutions. There is no resuspension of particles that have beached; the beached location is considered the final sink location.

The advection of particles depends on surface ocean currents taken from two different models which were used to evaluate the transport of particles and help quantify uncertainty in the results. The NEMO-based CMEMS Global Ocean Physics Analysis and Forecast hydrodynamic model (E.U. Copernicus Marine Service Information (CMEMS), Marine Data Store (MDS), 2022a) has a resolution of 1/12°, which is roughly 9.2 km at the latitudes of the Bay of Bengal, and includes data assimilation (Lellouche et al., 2018). Also included was the ROMS-based high-resolution model, configured for the North Indian Ocean as a part of the High-Resolution Operational Ocean Forecast and Reanalysis System (known as NIO-HOOFS) by INCOIS for the Indian Ocean (Francis et al., 2020), which has a much higher resolution of 1/48°, corresponding to approximately 2.3 km at these latitudes, but does not include data assimilation. Hereafter these experiments will be referred to as CMEMS and ROMS, respectively. As this study aimed to determine the pathways of floating marine macro-litter across the Bay of Bengal, only surface currents were required to drive such buoyant items. Following some sensitivity tests detailed in Appendix B, particles were forced with daily-mean ocean, Stokes drift, and wind velocities. Additional datasets from CMEMS Global Ocean Wave Analysis and Forecasting model (Ardhuin et al., 2010; E.U. Copernicus Marine Service Information (CMEMS). Marine Data Store (MDS), 2022b) and ERA5 global atmospheric reanalysis (Hersbach et al., 2023) were used to provide Stokes drift velocities and wind fields at a height of 10 m above land, respectively. Stokes drift velocities were available in 3-hourly timesteps at a resolution of 1/5°, which is roughly 21 km at the latitudes of the Bay of Bengal; wind velocities were hourly and with a resolution of 1/4°, which is approximately 26 km at these latitudes. Both datasets were interpolated onto the relevant grid for each of the CMEMS and ROMS runs using cubic interpolation and then averaged to daily timesteps.

Particle release locations were uniformly spaced around all major coastlines in the Bay of Bengal (Fig. 1). Particles were released on average 6 km from the Natural Earth coastline (naturalearthdata.com), with a maximum distance of 18 km in

some locations. This distance was chosen to complement different coastlines from the two hydrodynamic models, ensuring no particles were released on land while also ensuring they were released on the continental shelf for both configurations; this ensured coastal dynamics rather than open ocean dynamics influenced the particles when they initiated their journeys. We chose to release the particles from exactly the same latitudes and longitudes in both simulations, but note this means their proximity to the coast will differ between the CMEMS and ROMS runs due to the differences in hydrodynamic model resolution (Fig. 1c-d). A particle was released from each of the 500 coastal locations every day for a year, with 182,500 particles released in total. The number of released particles is consistent with other particle tracking studies conducted in the Bay of Bengal and on a global scale (e.g. Chassignet et al., 2021; Chenillat et al., 2021; Lebreton et al., 2012; van der Mheen et al., 2020a). These idealised particle release locations are unrelated to the magnitude of litter sources in the Bay of Bengal because of the uncertainties in measurements of mismanaged waste entering the oceans. Instead, particle sources and sinks can be used to investigate potential pathways of litter from all along the coastlines surrounding the Bay of Bengal and weightings could be applied as a post-process in the future, should source estimates become more accurate.

Model simulations covered 1st June 2018 – 30th September 2019 for each case (CMEMS and ROMS). This time frame was chosen due to the overlap in available data for each model and to enable a full year of particle release, plus a further 4-month season to allow time for those released later in the year to reach the shore. The short timescale over which the simulations were run means that degradation and subsequent sinking of macroplastics can be neglected, as a previous study found that less than 2% of plastics would degrade over the course of a year and most microplastics found in the ocean today were produced in the 1990s or earlier (Lebreton et al., 2019). A model time step of 15 minutes was used (following Delandmeter and van Sebille (2019)) and particle positions were output daily.

The results discussed below focus on where particles released from six countries (Sri Lanka, India, Bangladesh, Myanmar, Thailand, & Indonesia; Fig. 1) beach, to determine the final sinks and discuss country-country connections. The Andaman & Nicobar Islands do not release any particles during these simulations because the population density is so low that very little litter is expected to originate from there. These islands are a territory of India but were treated separately from mainland India when calculating connections between the coasts. Hereafter, any reference to India refers to mainland India in the Bay of Bengal only. The reversal of the winds and ocean currents during the year associated with the monsoon was expected to significantly impact some patterns of litter trajectories. We therefore ran separate simulations for each season, with particles released over a season-specific, four-month period: monsoon = 1st June – 30th September 2018; post-monsoon = 1st October 2018 – 31st January 2019; pre-monsoon = 1st February – 30th May 2019 (Anoop et al., 2015). Regardless of the release period, all particles were tracked until the end of September 2019.

**2.1 Model limitations**

While this particle tracking model includes the fundamental drivers of litter dispersal to paint a reliable picture of where floating litter released from a given country may finish its journey, there are processes neglected by the model that limit its ability to fully capture the behaviour of marine debris. These limitations are important to keep in mind while assessing the results of this study.

The beaching process is a critical step in the journey of a piece of marine litter (Hinata et al., 2020a) yet there is no consensus on how best to implement this step in particle tracking models. Some researchers have used a similar method to this study whereby a particle was deemed to be beached when its position was on a land grid cell (e.g. Irfan et al., 2024), whereas others considered particles beached if they persisted in a coastal grid cell for a given amount of time (e.g. Isobe and Iwasaki, 2022). Several other studies have taken the approach of probabilistic determination. For example, van der Mheen et al. (2020a) used a random probability to determine if a particle would beach, so long as it was within a given distance of the coast and that distance was decreasing. Chenillat et al. (2021) and Onink et al. (2021) used similar methods to this. Nevertheless, each of these methods used to determine particle beaching are simplistic and neglect much of the nuance involved with beaching processes in reality. Therefore, this study acknowledges the limitations of using this approach to determine particle beaching. Regardless of the beaching method employed, the resolution of all these hydrodynamic models, including the two used in this study, are too coarse to fully capture all processes that are key to marine debris beaching. Fine-scale ocean dynamics such as submesoscale and microscale eddies near the coast contribute to litter accreting and washing ashore but are not represented even in the finer scale ROMS model we used to advect particles. Sub-grid scale tidal motions at the shoreline are also precluded, yet they would likely lead to higher beaching rates (Zhang et al., 2020), and slope at the coastline is not represented by either model. Additionally, the shape of the coastline, while much more realistic in the ROMS model versus CMEMS (Fig. 1), is still not refined enough to show the true morphology of the coast and misses many features such as estuaries which have been demonstrated to act as traps for floating debris (Duncan et al., 2020; Pawlowicz et al., 2019). The model also omits the resuspension of particles once they have beached. If resuspension were to be included in the model, the final beached locations of particles may demonstrate differing connectivity and the proportion of particles left afloat would also be expected to increase. Pawlowicz et al. (2019) found in their observations of drifters in an estuary that if a drifter refloated after beaching, it was more likely to reach the open ocean. Thus far, there is little empirical evidence to suggest what resuspension rate should be used in a Lagrangian model (Hinata et al., 2020b), and such a rate might differ in different regions depending on the morphology of the coastline.

Stokes drift and windage are the main mechanisms driving beaching in the model, although there is no strategy applied to restrict beaching via other drivers such as diffusion, but there are some limitations of the datasets used and how they are implemented. While the ocean current velocity datasets, particularly the ROMS data, have relatively high spatial resolution for a regional model such as this, the Stokes drift and wind velocities used in both the CMEMS and ROMS simulations are coarser than the ocean velocities. Any differences in beaching between the two simulations is therefore expected to result from the differences in general circulation patterns as opposed to wind and wave effects. Another limitation to note in relation to the Stokes drift velocities implemented in the model is that the simple addition of these velocities to those of the ocean velocities does not provide accurate forcing for particles. This method neglects any feedbacks which might arise between ocean currents and wave induced movement that would affect both velocities. This would be remedied if using a coupled model. The use of only surface velocities rather than running a 3D simulation further limits the movement of particles.

However, despite the particle tracking simulations being limited to 2D, the hydrodynamic simulations were run in 3D, and this mitigates some of these shortcomings.

## 2.2 Validation using drifter trajectories

To assess model performance, the simulated trajectories of floating litter were compared with paths of drifters which had lost their drogues in the Bay of Bengal between June 2018 – September 2019. Undrogued drifters would float at the surface of the ocean and are therefore analogous to some types of floating marine litter. The movement of floating litter at the surface of the ocean differs due to factors such as shape and density, particularly with respect to the effect of wind (Pereiro et al., 2018). Therefore, drifters are not expected to represent all items of floating litter, but they are one of the closest analogies that can be tracked to validate the particle tracks in the model. Within the Global Drifter Program's quality-controlled 6-hour interpolated dataset (Lumpkin and Centurioni, 2019), five drifters were identified that met these criteria within the spatial and temporal limits of the model (Fig. 2a). Most of these drifters began and continued their journeys in the open ocean. Consequently, minimal proportions of these drifter trajectories can be considered to verify coastal dynamics. This is an important consideration given that the CMEMS simulations include data assimilation (for sea level, temperature and salinity) and would therefore be expected to provide more accurate offshore currents than the ROMS velocities. As the separation between the particles and drifter location is expected to increase with time (Tamtare et al., 2021), each drifter trajectory was separated into week-long segments. This ensures no bias in comparison based purely on the duration of drifter trajectory.

CMEMS and ROMS simulations were run using the same input data and parameters described for the main simulations. Starting at midday on the first full day after each drifter lost its drogue, 100 particles were released at the same location as the drifter. For each subsequent week, a further 100 particles were released from the location of the drifter at that time. Each particle was then followed for one week and compared to the relevant drifter trajectory during that time (Figs. 2b & C1). Note that the random-walk diffusion causes each of the 100 particles to take a slightly different path. To quantify model performance, the mean cumulative separation distance of weekly trajectories (MCSD$_{week}$) was calculated for all particles and corresponding drifter locations at each timestep, following Haza et al. (2019), van der Mheen et al. (2020b):

$$MCSD_{week} = \frac{1}{TP}\sum_{t=1}^{T}\sum_{p=1}^{P}|x_p(t) - x_d(t)|, \tag{1}$$

where $x_p(t)$ and $x_d(t)$ are the locations of the particle and drifter, respectively, at time $t$. $P$ is the total number of particles and $T$ is the total number of timesteps.

## 3 Results

### 3.1 Validation of trajectories

The MCSD$_{week}$ across all five drifters was 66 km for the CMEMS run and 92 km for the ROMS run (Fig. 2c). The lowest MCSD$_{week}$ in the CMEMS run was for Drifter 5 (D5, 38 km), while the highest MCSD$_{week}$ was for Drifter 4 (D4, 71 km). For the ROMS run, the lowest MCSD$_{week}$ was associated with Drifter 1 (D1, 44 km) and the highest MCSD$_{week}$ with Drifter 3 (D3, 144 km).

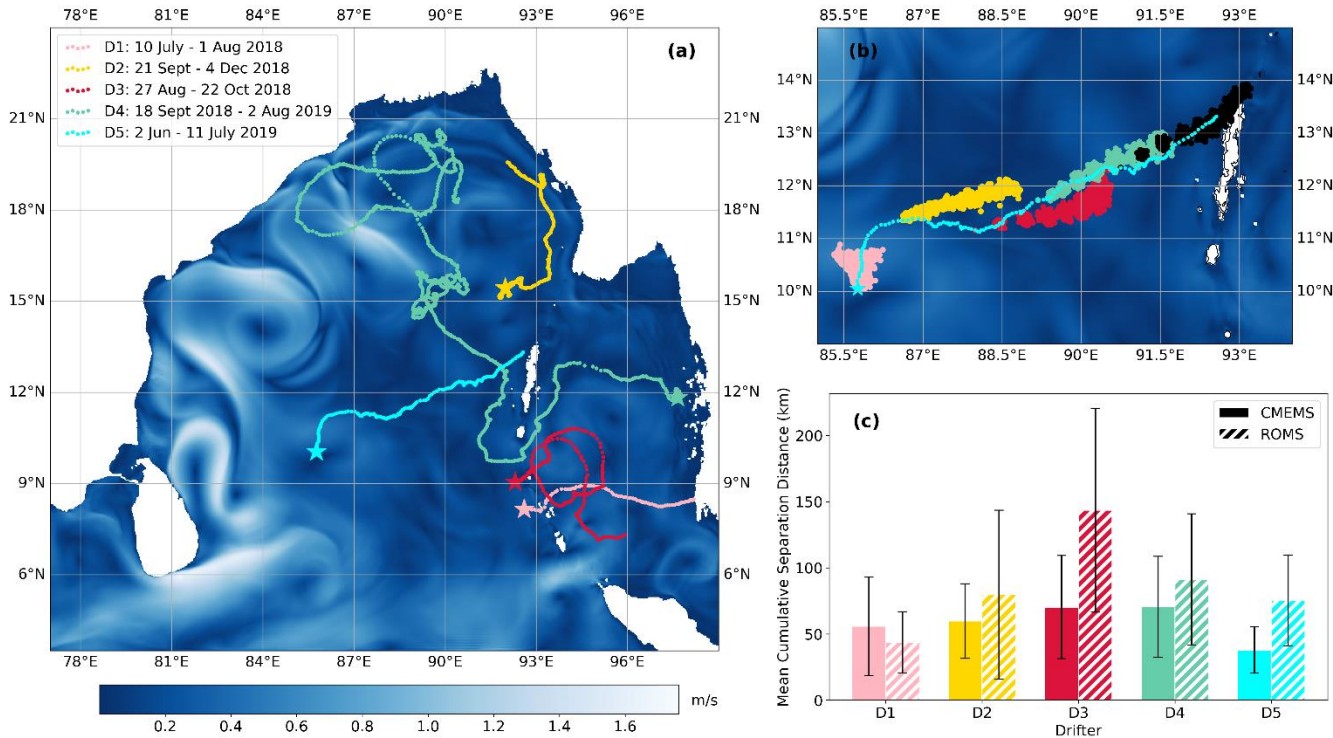

**Figure 2: (a) Tracks of undrogued surface drifters used to validate the paths of particles in the Bay of Bengal during the simulation time-period. Stars indicate the starting position of each drifter, corresponding to the day after it lost its drogue. Background depicts a snapshot of ocean current speeds from the ROMS dataset from 2nd June 2019. (b) Particles from the ROMS simulation released once per week at the location of the drifter at that time are colour-coded to show how closely they follow the D5 drifter track, shown in cyan. Background shows ocean current speeds from the ROMS dataset from 2nd June 2019. Other drifter tracks with corresponding validation particles for both CMEMS and ROMS runs are shown in Appendix C. (c) Mean Cumulative Separation Distances for each drifter in both the CMEMS (solid colour) and ROMS (striped colour) simulations. Error bars depict the standard deviation.**

## 3.2 Annual connectivity

Of the 182,500 particles released throughout the year, both the CMEMS and ROMS experiments showed the vast majority beached along the Bay of Bengal coastline within 16 months (83% and 91%, respectively; Table 1). While the number left afloat was higher at the end of the CMEMS experiment, in both cases, this was a very small proportion of the total particles (<0.5%). Particles that did not beach were predominantly found to leave the domain through open boundaries. This proportion was higher for CMEMS than ROMS (16% versus 9%, respectively), with approximately half the total escaped particles being lost through the southwest boundary in both cases (Tables D1-6).

The majority of particles were found to beach on their country of origin; at least 57% in the CMEMS run and 69% for the ROMS run (Fig. 3a-b). The second highest connectivity rate from almost every country was towards Myanmar, up to 29% (CMEMS) and 14% (ROMS). Notably, there is relatively low connectivity (≤2%) towards Thailand or Indonesia from

|  | Full Simulation (Jun '18 – Sept '19) | | Monsoon (Jun – Sept) | | Post-monsoon (Oct – Jan) | | Pre-monsoon (Feb – May) | |
|---|---|---|---|---|---|---|---|---|
|  | CMEMS | ROMS | CMEMS | ROMS | CMEMS | ROMS | CMEMS | ROMS |
| Beached | 83% | 91% | 87% | 95% | 73% | 85% | 91% | 94% |
| Left domain | 16% | 9% | 13% | 5% | 27% | 15% | 8% | 6% |
| Remained afloat | <0.5% | <0.5% | <0.5% | <0.5% | <0.5% | <0.5% | <0.5% | <0.5% |

**Table 1: Seasonal breakdown of particles that beached along coastlines in the domain, left the domain through an open boundary, or were still afloat at the end of the simulations.**

any of the other four countries. The main difference between CMEMS and ROMS results is that particles in the ROMS run
were less dispersed; generally, a higher fraction of particles released from a given country beached on their own shores and a lower fraction beached on neighbouring countries.

**3.3 Seasonal variations**

The following results describe the fate of particles released during each season separately, i.e. "monsoon" particles refer to particles that were released between 1st May – 30th September 2018, regardless of when they settled.

**3.3.1 Monsoon**

In total, 87% of particles in the CMEMS run and 95% of particles in the ROMS run, released during the monsoon season, beached somewhere in the domain (Table 1). Almost all the remaining particles left the domain without beaching (CMEMS: 13%, ROMS: 5%). The majority left through the southwestern boundary, towards the Arabian Sea, in the CMEMS simulation (6%), but through the southern boundary, to the southern Indian Ocean, in the ROMS run (2%). This is the only season with
260 conflicting results in terms of exit locations for the different experiments. The direction of the different currents in each model which led to these differences can be seen in the Supplementary Animations.

Most particles released beached on their country of origin, for both the CMEMS and ROMS simulations (Fig. 3c-d). In the CMEMS run, the second highest beaching rate was always on a country in the anticlockwise direction, except for Sri Lanka, whose next highest connectivity rate was towards Myanmar (24%). The pattern was the same for the ROMS run apart
from particles released from India beaching on Myanmar with the second highest rate (16%). When considering only particles that were released and beached in the monsoon season, this pattern is less pronounced for both cases, with higher proportions of particles released from Sri Lanka, India and Bangladesh beaching on counties in a clockwise direction (Fig. 4a-b).

### 3.3.2 Post-monsoon

Only 73% (CMEMS) and 85% (ROMS) of the total number of particles that were released during the post-monsoon season beached throughout the simulations (Table 1). While still a large proportion, these figures are noticeably lower than the other two seasons. The number of remaining particles leaving the domain was also higher than the monsoon and pre-monsoon seasons combined (CMEMS: 27%, ROMS: 15%) by a substantial margin, making the particle-exit pattern from this post-monsoon season dominant across the full simulation for the CMEMS and ROMS cases. The majority of these particles left the domain through the southwestern boundary towards the Arabian Sea (CMEMS: 19%, ROMS: 10%), a smaller portion leaving through the southern open boundary (CMEMS: 7%, ROMS: 3%), and relatively few leaving through the southeastern boundary into the Strait of Malacca (1%) in each case. These groupings are reflected in the particle-exit pattern for the full simulation in both cases.

This is the only season where own-country beachings do not represent the greatest connectivity for all countries in the CMEMS run (Fig. 3e). For the ROMS simulation, while own-country beaching rates are always highest (Fig. 3f), rates were significantly lower for Bangladesh, Myanmar and Thailand than for other seasons, or the year overall. There were significant differences in the spread of particles from Bangladesh and Thailand for each simulation. In the CMEMS run, more particles beach on Bangladesh's neighbours, Myanmar (55%) and India (21%), than Bangladesh itself (15%), whereas in the ROMS run, own-country beach remained highest for Bangladesh (44%), with smaller but still significant rates reaching Myanmar (28%) and India (17%). For particles released from Thailand, only 13% of particles beach on their own coastline in the CMEMS run, with more particles beaching on Myanmar (28%), Sri Lanka (24%), India (15%), and Andaman & Nicobar (14%). In contrast, the ROMS run shows 60% of particles released from Thailand also beached there, and only Andaman & Nicobar received more than 10% of the remaining portion. For both experiments, almost no particles released from Sri Lanka reach the eastern Bay of Bengal, beaching predominantly along India or its own shores. In fact, a very large proportion of particles released from Sri Lanka in the post-monsoon season left the domain: 64% in the CMEMS run and 39% in the ROMS simulation (Tables D3&4).

### 3.3.3 Pre-monsoon

For particles released during the pre-monsoon period, 91% in the CMEMS run and 94% in the ROMS run beached within the domain before the end of the simulation (Table 1). Of the 60,000 particles released in this season, 8% in the CMEMS run and 6% in the ROMS run left the domain, predominantly via the southeastern boundary, towards the Strait of Malacca (CMEMS: 4%, ROMS: 2%).

For both experiments, beaching rates were highest for particles settling on their country of origin (Fig. 3g-h). The next highest proportion was often found on Myanmar, aside from particles released from India and Indonesia, where particles beached mostly on Bangladesh (CMEMS: 10%, ROMS: 2%) and Thailand (6% in the ROMS run), respectively. Both the CMEMS and ROMS experiments show noticeably fewer particles spreading from east to west during the pre-monsoon season, particularly when compared with the post-monsoon season.

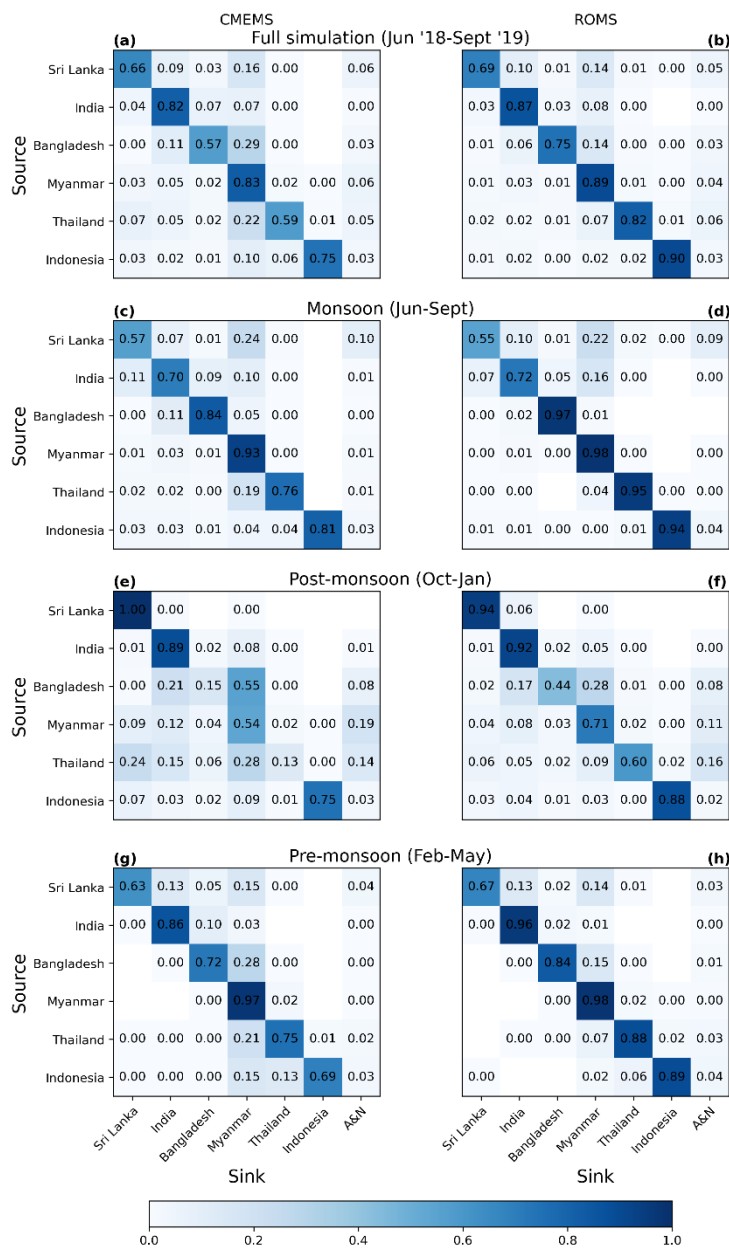

**Figure 3: Connectivity matrices showing sources and sinks of particles normalised by the number of particles released from each country. Left column shows results from the CMEMS simulations and the right column shows those of the ROMS runs. Top row shows connectivity over the course of the full simulation (a-b); second row shows results from particles released during the monsoon season (June – September 2018; c-d); third row shows connectivity of particles released during the post-monsoon season (October 2018 – January 2019; e-f); and the bottom row shows results of particles released in the pre-monsoon season (February – May 2019; g-h). Blank boxes show where no particles have connected between countries; boxes showing "0.0" have been rounded down but are in fact a non-zero value.**

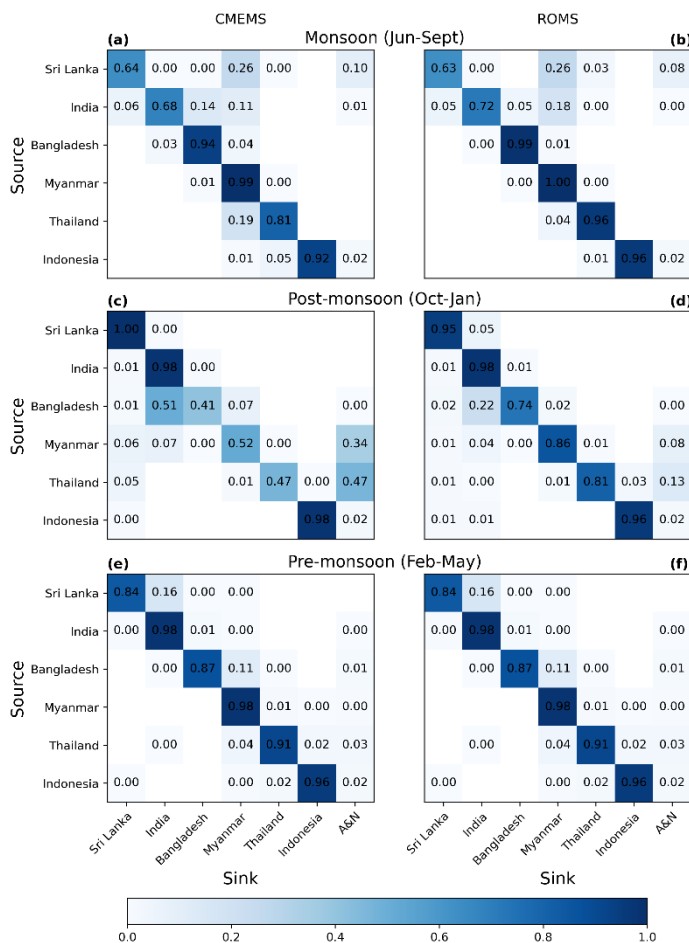

**Figure 4: Connectivity between source and sink locations made by particles that were released AND beached during the specified period of time: monsoon period only (June – September 2018; top row), post-monsoon season only (October 2018 – January 2019; middle row), and pre-monsoon season only (February – May 2019; bottom row). Results from the CMEMS run are shown in the left panel and the ROMS run on the right. Blank boxes show where no particles have connected between countries; boxes showing "0.0" have been rounded down but are in fact a non-zero value.**

## 4 Discussion

### 4.1 Connectivity between countries

While reporting connections based on the country of release might overlook issues such as shared watersheds, it also has the advantage of making our results directly comparable between studies and quantifying progress once measures are taken. Intentionally, the litter sources were all normalised in this study, rather than providing numbers of mass export between countries, so that the focus is on the efficiency of the marine pathways. Where results from the CMEMS and ROMS models agree, we have confidence in conclusions drawn about country-to-country connections within the Bay of Bengal, whereas instances where the models differ can point to uncertainties in connectivity. For each experiment, own-country beaching

showed the predominantly highest rates for all particles released. This is consistent with previous modelling studies in the region (e.g. Chassignet et al., 2021; Chenillat et al., 2021) and is unsurprising given that the resolution of global or regional-scale models is insufficient at the coast to implement realistic beaching processes and instead, simpler beaching methods are employed in this study and others.

Few modelling studies have quantified the connections between marine litter released from a given country and where that litter lands. van der Mheen et al., (2020a) calculated connections between countries but chose to publish their results as the percentage of total particles that beached on a given country rather than total particles released from a given country. Therefore, our results are not comparable with their findings. However, Chassignet et al. (2021) used their global model to detail such connections during a 10-year simulation, allowing a comparison with our results from the Bay of Bengal. A direct comparison of proportions of particle beachings cannot be made for India, Thailand, and Indonesia, since all have coastlines which have been excluded from our study but are included in the global configuration of Chassignet et al. (2021). However, this would not affect the rankings of which countries have had the greatest number of modelled beachings on the other five countries within the Bay of Bengal. The main difference between our model and that of Chassignet et al. (2021), other than the forcing data used to advect the particles, is the release locations of litter. Chassignet et al. (2021) used both inland river and coastal input locations based on mismanaged waste estimates from Lebreton et al. (2017) and Lebreton and Andrady (2019) rather than the uniform release approach we have taken here. Therefore, the relative fractions transported between countries will be dependent on the assumptions made around these sources.

Chassignet et al. (2021) found that own-country beachings were highest for all six countries included in our model, consistent with our study. They determined that the second highest beaching rate for litter released from India, Bangladesh and Myanmar was on the same country as found in the current study (Myanmar, Myanmar and India, respectively. Note that the second greatest fraction of particles released from Myanmar in our simulations beached on Andaman & Nicobar, which are Indian islands, with the next highest fraction beaching on mainland India). However, differences were seen in the second highest beaching rates of litter released from Sri Lanka (Chassignet et al. (2021): India, this study: Myanmar) and Thailand (Chassignet et al. (2021): Indonesia, this study: Myanmar). Beaching rates of particles released from Indonesia in our simulations differed between runs. The CMEMS run showed the second highest beaching rate for particles released from Indonesia was on Myanmar, whereas in the ROMS run, particles were found to beach on Andaman & Nicobar (India) with the second highest rate, which is consistent with the findings of Chassignet et al. (2021).

Relatively high rates of particles ($\geq$7%) released from all countries in our simulations were found to beach on Myanmar's shores, except Indonesia in the ROMS run. The relative size of the Myanmar coastline compared with other countries in the Bay of Bengal, combined with the fact that the EICC flows towards Myanmar for a large proportion of the year (monsoon and pre-monsoon seasons), could account for this. Chassignet et al. (2021) also found significant proportions of litter from the five other countries in the Bay of Bengal beached on Myanmar (1.5-24.1%), making it the country that received the greatest proportion of litter from the other five countries within the Bay of Bengal.

## 4.2 Monsoonal influence on marine litter pathways

Seasonal variability in beaching rates was influenced by wind and ocean currents, with countries on the eastern side of the Bay of Bengal (Bangladesh, Myanmar and Thailand) having high own-country beaching rates during the monsoon (CMEMS: 76-93%, ROMS: 95-98%) and pre-monsoon seasons (CMEMS: 72-97%, ROMS: 84-98%), as opposed to lower rates during the post-monsoon season (CMEMS: 13-54%, ROMS: 44-71%). The opposite pattern is found for countries on the western side of the Bay of Bengal (Sri Lanka and India), with relatively low rates during the monsoon (CMEMS: 57-70%, ROMS: 55-72%), which get higher over the post-monsoon period (CMEMS: 89-100%, ROMS: 92-94%), before Sri Lanka's own-country beaching rates drops back down in the pre-monsoon season (CMEMS: 63%, ROMS: 67%), while India's rates remain high (CMEMS: 86%, ROMS: 96%). This is a result of north-eastward flowing monsoon and pre-monsoon currents transporting litter away from countries in the west and towards countries in the east (Fig. 1). During the post-monsoon season, the main current (EICC) flows south-westward, reversing this transport of litter (see Supplementary Animations).

Transport of particles released in the pre-monsoon season is consistent with the direction of the EICC and winds during this season (Fig. 1), with few particles beaching on countries to the west or south from India, Bangladesh or Myanmar, in either scenario. Instead, almost all particles beach on their country of origin or a country to the north or east. This pattern is less pronounced for the monsoon season. Surface currents in the Bay of Bengal are stronger in the pre-monsoon than in the monsoon season, but winds are stronger during the monsoon (Phillips et al., 2021). Therefore, the differences between these two seasons may result from relative influence shifting between ocean-current advection as opposed to windage or Stokes drift.

The seasonal analysis considered here separates particles based on the time of release. However, particles may remain afloat for longer than their source season (Figs. 5&6), and therefore particles that did not beach during the season they were released would be affected by changes in current or wind direction later in their trajectory. Monsoon particles are defined here as particles released during 1st June – 30th September 2018, but any particles that remain afloat after the monsoon season may then be influenced by post-monsoon winds and currents. A particle released, for example, during the post-monsoon season that did not beach until the pre-monsoon season might have been influenced by the south-westward flowing EICC during the post-monsoon period and then carried in the opposite direction by the reversal of this current in the pre-monsoon season. In contrast, the pre-monsoon particles are advected for a further four months of monsoon currents after their initial release, before the end of the simulation, so would not be expected to show this transport towards the south/west. Figure 5 shows that the proportion of particles released in each season that beached in a later season was sizeable, particularly for particles released during the post-monsoon period. This could affect our interpretation of results. Therefore, to investigate any bias in beaching patterns resulting from particles that did not beach within the season they were released, we calculated the connectivity of particles that were released *and* beached with the four-month window of each season (Fig. 4). The patterns of beaching toward the south/west in the monsoon season is reduced in this case. The connectivity matrices show expected patterns of beaching for Sri Lanka, India, Bangladesh and Myanmar, which are all in the path of the EICC, as greater rates of beaching are seen on

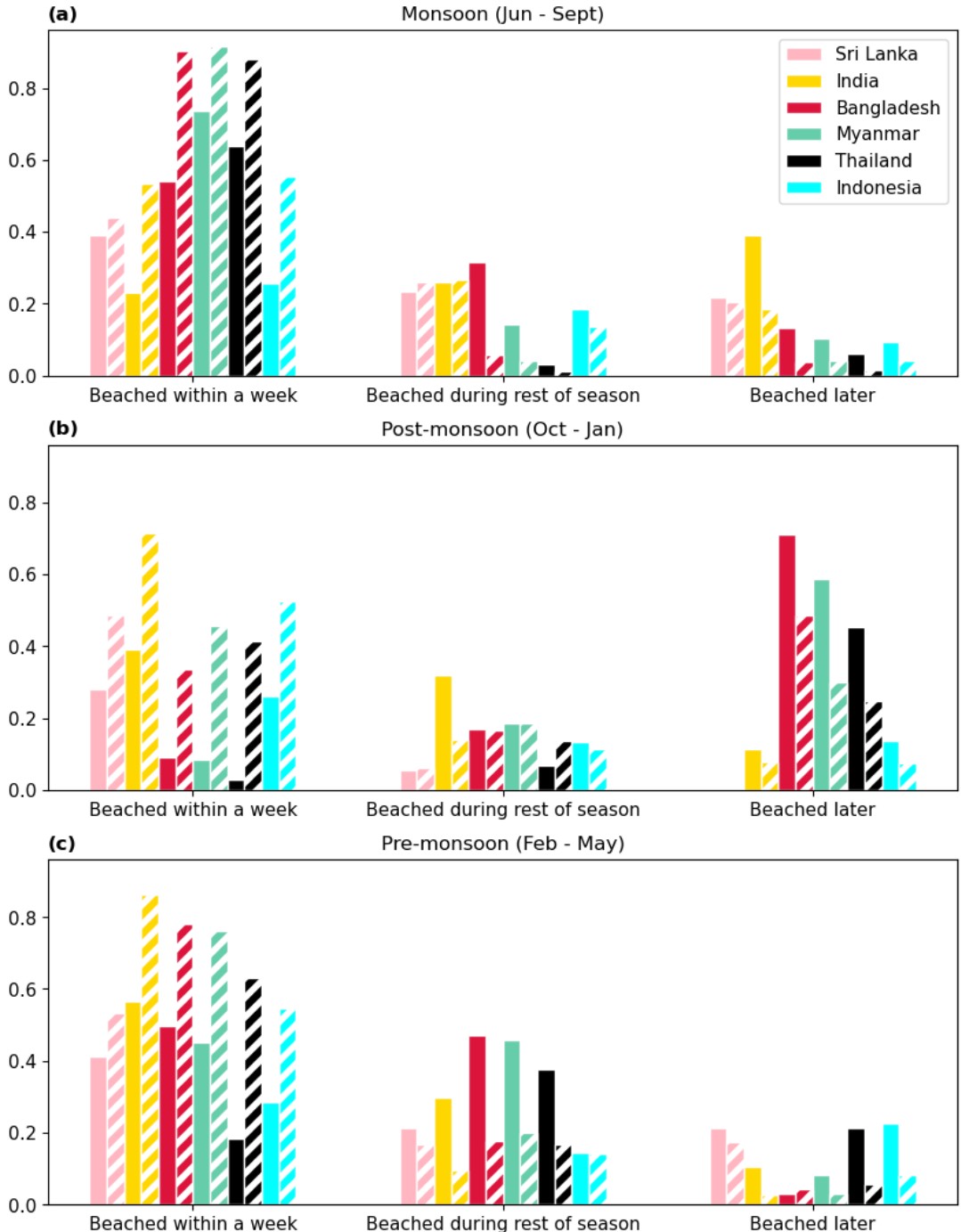

**Figure 5: Fraction of particles released during the monsoon season (a), post-monsoon season (b) and pre-monsoon season (c) from each country that beached within given timeframes. Bars are colour-coded by country with solid bars representing CMEMS particles and striped bars representing ROMS.**

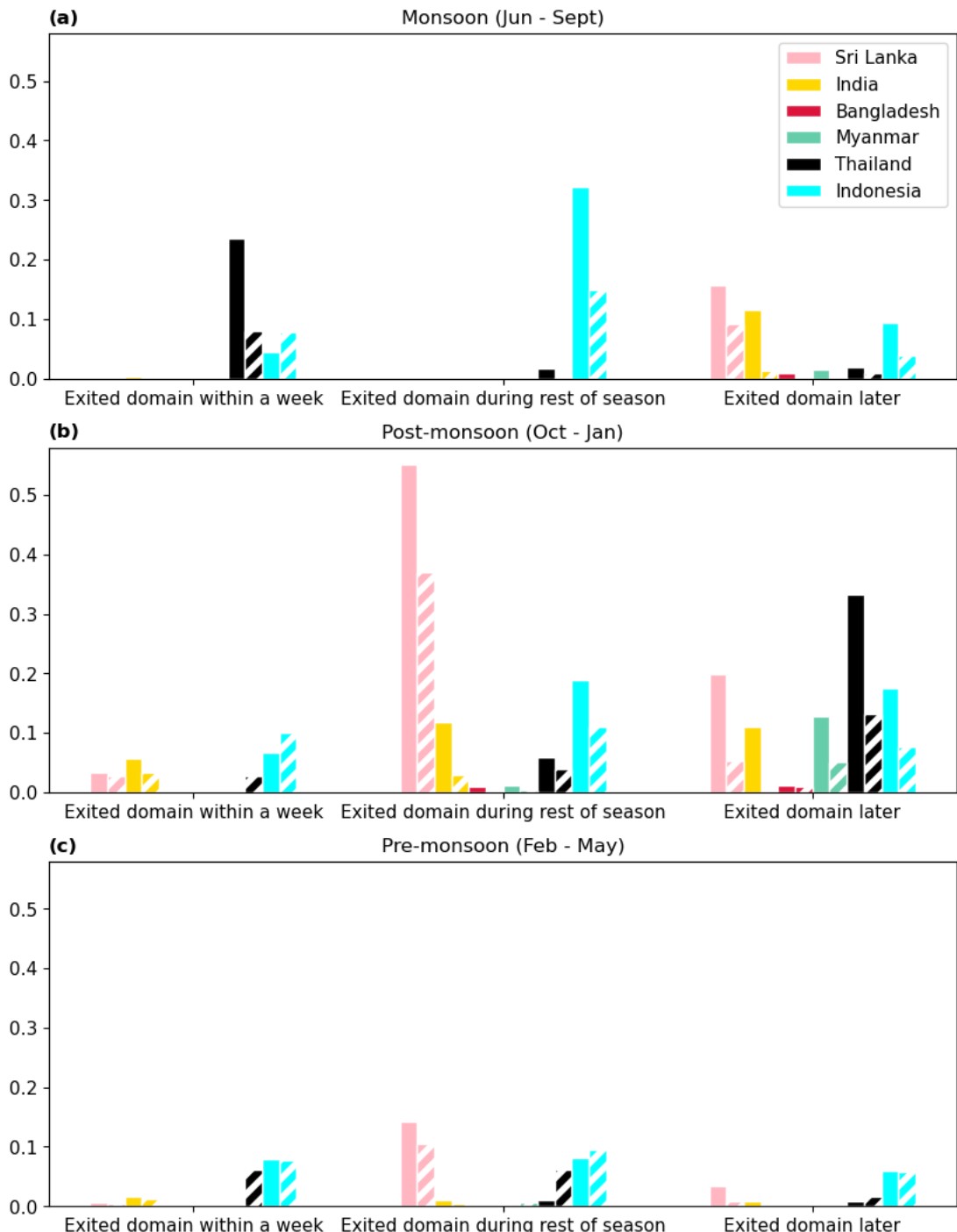

**Figure 6: Fraction of particles released during the monsoon season (a), post-monsoon season (b) and pre-monsoon season (c) from each country that left the domain within given timeframes. Bars are colour-coded by country with solid bars representing CMEMS particles and striped bars representing ROMS.**

countries to the north and east during the monsoon and pre-monsoon seasons and lower rates to the south and west. This pattern is reversed for the post-monsoon season. Therefore, the duration of the trajectories likely explains some of the unexpected results seen in Fig. 3.

The expected pattern of most particles from Myanmar and Bangladesh settling on Andaman & Nicobar, Sri Lanka and India in the post-monsoon season due to the south-westward EICC and winds is also not clear from the connectivity matrices (Fig. 3e-f), although greater proportions of particles being released from Bangladesh and Myanmar are beaching on countries to the south/west in this season compared with those from the pre-monsoon and monsoon periods. Post-monsoon-released particles that do not beach within that season are given a further eight months to beach which means they are subjected to pre-monsoon and monsoon currents that would propel them north-eastward. Some of the high proportions of particles released in the post-monsoon period that beached in a surprising direction (e.g. Bangladesh->Myanmar) can therefore be explained because a large percentage of particles were found to beach after this season (Fig. 5). The unexpected pattern of beaching behaviour disappears when only accounting for particles that were released and beached within the post-monsoon season (Fig. 4c-d). Indonesia, which is not in the path of the reversing EICC, does not show a significant seasonal pattern; own-country beaching rates remain relatively steady throughout the year (CMEMS: 69-81%, ROMS: 88-94%). The Northeast and Southwest Monsoon Currents travel across the Bay of Bengal between the southern tip of Sri Lanka and Indonesia and might be expected to transport litter between these two countries. However, while a significant proportion of particles travel from Indonesia to beach on Sri Lanka, particularly in the post-monsoon season when the Northeast Monsoon Current would be expected to transport particles in this direction, strikingly few particles make the journey from Sri Lanka to Indonesia. Almost all particles from Sri Lanka that eventually end up on a coast either beach on Sri Lanka itself or are carried northeast by the EICC (see Supplementary Animations). More particles released from Indonesia are caught in local eddies and transported north, beaching on Thailand, Myanmar, and Andaman & Nicobar, particularly in the pre-monsoon and monsoon seasons. These findings indicate that the EICC exerts more influence over particle trajectories in the Bay of Bengal than the other major currents in this region.

The majority of particles that left the domain, throughout both the CMEMS and ROMS runs, did so through the southwestern boundary towards the Arabian Sea during the post-monsoon season (CMEMS: 19%, ROMS: 10%), with particles released from Sri Lanka making up the largest contingent of this exodus (Tables D3&4). A substantial percentage of post-monsoon particles from Thailand, India, Myanmar and Indonesia also left the domain through this boundary in both models, flowing either south-westward with the EICC or westward with the Northwest Monsoon Current. This is in line with van der Mheen et al. (2020a) who found particles in their own simulations of the wider northern Indian Ocean were transported from the Bay of Bengal into the Arabian Sea during December – February. During our post-monsoon season, a portion of particles also escape to the southern Indian Ocean (CMEMS: 7%, ROMS: 3%), mostly from Thailand and Indonesia, again consistent with van der Mheen et al. (2020a) who found that up to 5% of particles crossed the equator into the southern Indian Ocean in the period September – November. This southward export during the post-monsoon season would be consistent with the southward direction of the EICC at this time of the year. In contrast, the monsoon and pre-monsoon seasons, during which

ocean currents in the Bay of Bengal primarily travel north-eastwards, would not facilitate such an export from the model domain.

## 4.3 Model comparisons

Although the CMEMS and ROMS runs do show many of the same patterns, which gives us confidence in the validity of the results, there are significant differences which indicate one model may be performing better than the other in certain regions of the Bay of Bengal. Considerably more litter beaches on the country of origin in the ROMS run, and therefore rates of particles beaching elsewhere are noticeably higher in the CMEMS run for many connections (up to 29% vs 14%). In addition, markedly more particles beach soon after release in the ROMS run compared with the CMEMS simulation, in all seasons (Fig. 5).

The ROMS forcing dataset has much higher resolution than CMEMS, while CMEMS has data assimilation that ROMS does not include. Given the higher resolution of the ROMS data, we might expect that this model would more correctly represent slower flow along the coast. The inclusion of shallower cells would have a greater effect of seabed friction on the vertical shear stress; also, better-resolved coastal eddies would remove energy from the system with which to transport particles offshore. An example of the difference in the level of detail of coastal currents and mesoscale eddies resolved by each of the models can be seen in Fig. 1c-d. Slower coastal flow would increase the opportunity for diffusion to lead to beaching, accounting for higher coastal retention, or less particle dispersion, in the ROMS run. These mechanisms may also explain why more particles in the CMEMS run leave the domain compared with their ROMS counterparts (Table 1). However, the CMEMS data assimilation includes satellite sea surface height observations and therefore is expected to improve representation of currents offshore, where the ROMS currents are reported to have been underestimated (Sj et al., 2022). This appears to be verified by smaller $MCSD_{week}$ errors in the validation with drifters that were predominantly floating in the open ocean, away from the coast.

The largest difference in connectivity between the two simulations surrounds particles released from Thailand, with those from Bangladesh and Indonesia also showing substantial differences in the rates of own-country beachings versus beachings on other countries. The major ocean and wind currents in the Bay of Bengal essentially flow between Sri Lanka/India and Myanmar and both models capture own-country beaching rates on these three countries at a similar rate. Thailand and Indonesia are not directly in the path of the EICC which could explain discrepancies in the connectivity results as the coastal currents in this region of the Bay of Bengal are not driven by this major flow. The currents in this region may therefore be quite different in each model and the higher resolution of ROMS might indicate that the higher own-country beaching rates in this simulation are more indicative of real-life litter transport near the coast. Without further observations to provide validation of model performance, these differences highlight a degree of uncertainty in our results. Although the general patterns of country-country connections are similar in each case, the differences in magnitude of some connections demonstrates the importance of model resolution for accurately simulating coastal retention. These results highlight the requirement for a data assimilated model which can resolve small scale variabilities to force particle tracking simulations, which should perform well both along the coast and across the open ocean.

The majority of litter released in our simulations beached within the 16-month time period (CMEMS: 83%, ROMS: 91%), with most of the remainder leaving the domain through open boundaries rather than remaining afloat. van der Mheen et al. (2020a) found almost all litter released in the wider Indian Ocean beached within a few years, with countries in the Bay of Bengal being more heavily impacted than coastlines in the Arabian Sea. Our model accounts for several processes not included in van der Mheen et al. (2020a). Their model did not feature key mechanisms thought to promote the beaching of floating particles, such as windage or Stokes drift, instead assuming a beaching probability. Winds and waves are likely to have a large effect on beaching probabilities; Stokes drift, for example, has been found to reduce the residence time of particles in simulations in the Black Sea as well as increasing beaching rates by up to 75% (Castro-Rosero et al., 2023). Onink et al. (2021) found that not including Stokes drift in their global model reduced the trapping of particles near the coast and reduced beaching by 6-7%. Additionally, Irfan et al. (2024) found increases in beaching rates of 5% when Stokes drift was added to their model and a further 9% when windage was included. It is important to note, however, that neither model used in this study is able to fully resolve all the coastal processes that are likely to influence beaching rates.

We chose to release particles uniformly from the coastlines of all countries in our domain, with the exception of Andaman & Nicobar because of the relatively small population of this island chain. This decision resulted from the very large uncertainties associated with estimates of litter volume entering the oceans due to the paucity of measurements of waste generation (Jambeck et al., 2015). The sources and deposition of marine litter are also poorly constrained (Lebreton et al., 2017) with many researchers only accounting for major rivers as a conduit to the ocean (e.g. Irfan et al., 2024; van der Mheen et al., 2020a). Evidence has recently been found to suggest that smaller rivers may contribute more to marine litter if they traverse through an urban centre (Meijer et al., 2021). Additionally, Chenillat et al. (2021) found better agreement with the global distribution of floating marine litter when using particle release locations based on population density rather than river outflow locations. The researchers, therefore, stressed the importance of accurate litter inputs into particle tracking models for more trustworthy results. Releasing particles at river mouth locations based on estimates with such large uncertainties would compound uncertainties imposed by the model which would in turn lead to low confidence in the conclusions that have been drawn from our results. Instead, we opted to simply release particles uniformly along the entire coastline of the Bay of Bengal and look at normalised results based on the total number of particles released from a given country's coastline. Using this normalised connectivity, our results provide a first order approximation of sources and sinks within the Bay of Bengal region at different times of the year.

**5 Conclusions**

Two particle tracking simulations of floating marine macro-litter in the Bay of Bengal forced with different ocean velocity data showed some general trends from which we can draw conclusions. The majority of particles beached on their country of origin throughout the year, but the rates changed depending on the direction of the monsoon winds and associated ocean currents at different times of the year. Prior to and during the monsoon season, countries on the eastern side of the Bay of Bengal had higher rates of own-country beachings resulting from north-eastward flowing ocean velocities, while countries in

the west saw smaller proportions of their own particles beaching on their shores. In the post-monsoon season, this trend reversed along with the currents. The EICC, which flows between Sri Lanka/India and Myanmar (Fig. 1), appears to be the most influential current within the Bay of Bengal in terms of connecting particle sources and sinks.

Our results highlight that Myanmar received a significant number of particles from almost every other country. This is likely due to a combination of factors including the long length of Myanmar's coastline on which floating particles can beach, as well as the direction of EICC transport, towards Myanmar, for approximately two-thirds of the year. Due to the idealised nature of particle release locations in our simulations, we cannot quantify the amount of litter that would be expected to beach on Myanmar's shores. However, our results do allow us to quantify the efficiency of oceanic pathways within the Bay of Bengal that could facilitate transport of marine litter towards Myanmar.

Validation of our particle tracking model using undrogued surface drifters offshore showed a lower resolution hydrodynamic forcing dataset which incorporated data assimilation could better represent particle trajectories than a higher resolution model which did not include data assimilation. However, more observations would be required to further validate the results presented here in coastal locations. Differences in model results concerning the dispersion of particles from a given country suggest that model resolution may influence the transport of particles close to the coast. This study therefore demonstrates that both data assimilation and higher model resolution are required to accurately simulate the fate of coastal floating litter.

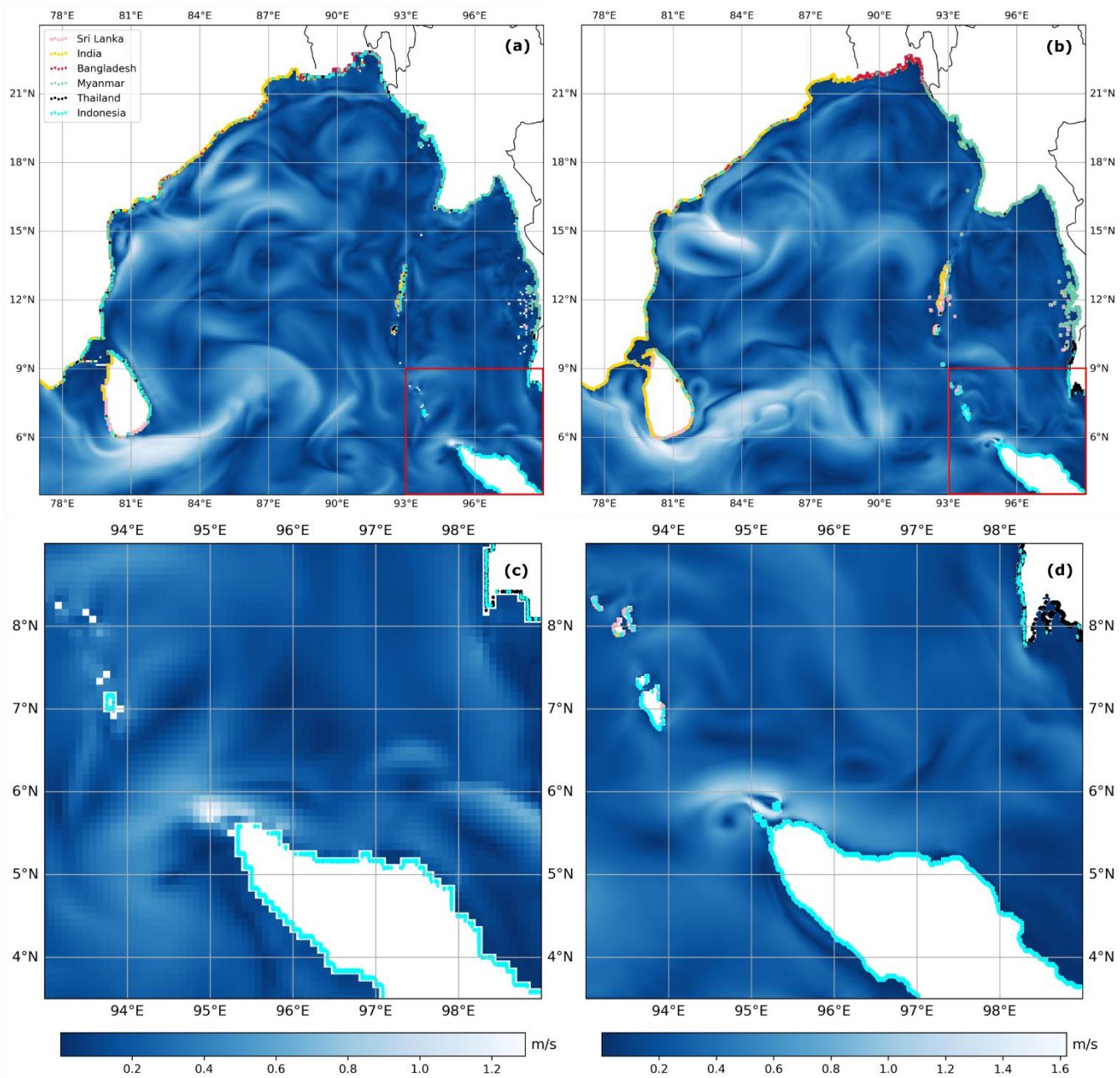

**Figure A1: Final locations of all particles that were considered to be beached, colour-coded by country of origin, in the CMEMS run (a) and ROMS run (b). Blue backgrounds show ocean current speeds on 24th July 2018 from each model and the areas in the red box are enlarged in (c) and (d), respectively.**

**Appendix B: Temporal resolution sensitivity tests**

To decide the required temporal resolution necessary to simulate particle trajectories across the Bay of Bengal, simulations were run to test the sensitivity of sink locations to temporal forcing. Simulations were forced with either CMEMS or ROMS hydrodynamic forcing (see main text for details) at either hourly or daily temporal resolution. All four simulations used the same parameters as well as wind and Stokes drift data as detailed in the main text and were run for the month of July 2020, with particles released every hour for the first two weeks only.

The results discussed in the main text focus mainly on where the particles beach, to determine the final sinks for the particles and discuss country-country connections. The runs used for these sensitivity tests were too short for many particles to beach, especially those travelling across the Bay of Bengal from the southwest to the northeast with the monsoon currents. Therefore, these results should be viewed solely with a view to establishing which resolution is required for a longer simulation that can determine the sources and sinks within the Bay of Bengal. The patterns of particles seen in the ocean at the end of the month-long runs show the similarities between the hourly and daily runs (Fig. B1a-d). To quantify any differences and determine if they are significant enough to warrant using the higher temporal resolution data, connectivity matrices were calculated for particles that did beach for each of the runs and the difference between the hourly and daily results was subtracted (Fig. B1e-f).

There are small differences in the beaching locations of particles in the hourly and daily resolution simulations when comparing CMEMS-forced runs (Fig. B1e). The largest difference is in the beaching locations of particles originating from Bangladesh. In the hourly run, 6% more particles released from Bangladesh beach on Bangladesh's coastline than in the daily run, where more particles end up on the shores of Myanmar. Differences between hourly and daily forcing using the ROMS model output were vanishingly small (Fig. B1f). Considering the aim of this study was to quantify connections between just six countries in such a large domain, as opposed to a more granular breakdown of the region, the differences in sink locations between the hourly and daily runs for each case (CMEMS and ROMS) were found to be negligible. Therefore, daily forcing was determined to be adequate to provide accurate results for the final experiment.

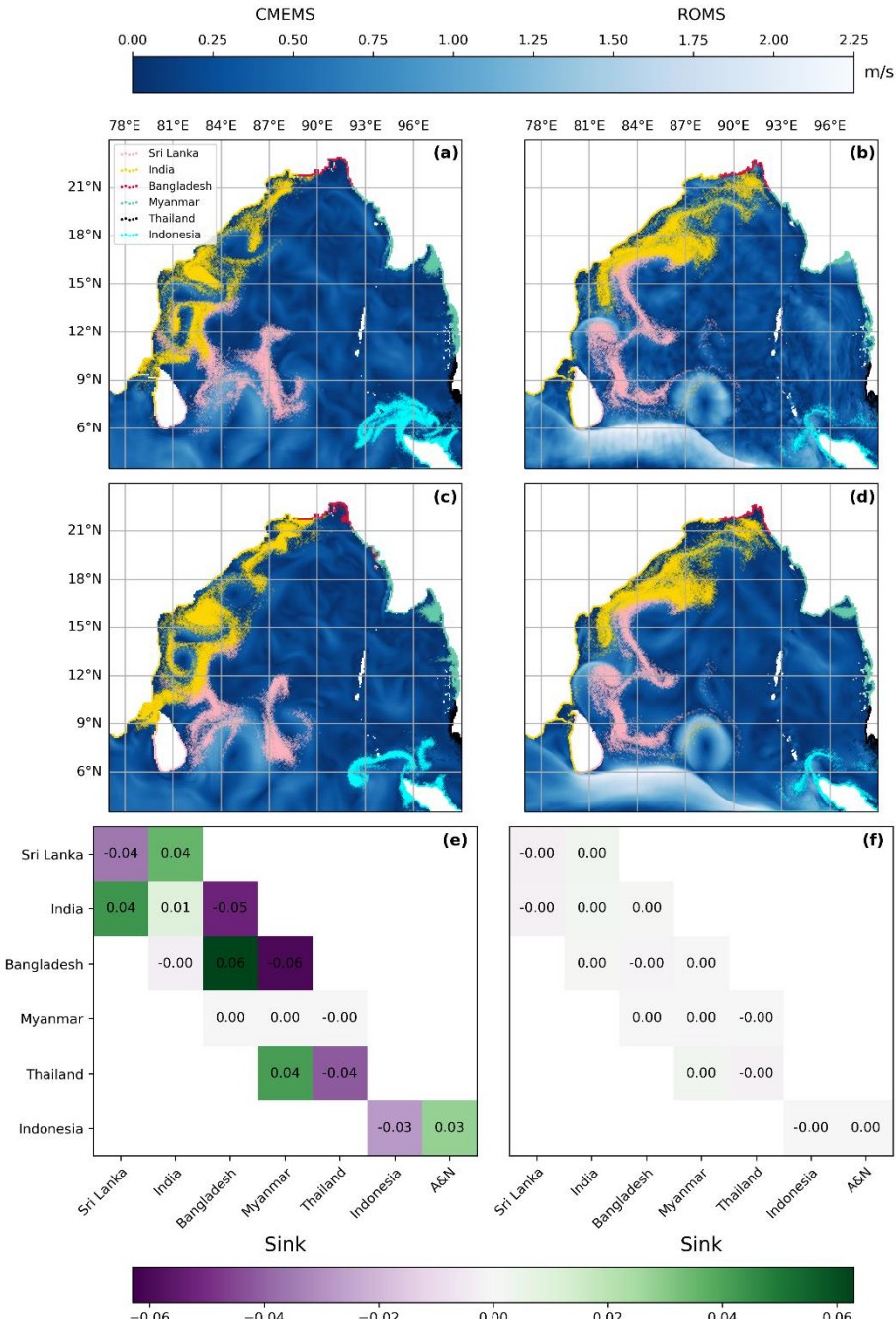

Figure B1: Particle positions at the end of an hourly-forced, month-long CMEMS run (a) and the corresponding hourly ROMS run (b) versus their positions at the end of the equivalent daily-forced CMEMS (c) and ROMS (d) simulations. Backgrounds show a snapshot of ocean current speeds on 31st July 2020 from the respective datasets. Connectivity matrices quantify the differences between CMEMS hourly and daily runs (e) and the hourly and daily ROMS runs (f). Only particles that beached during the month-long run in July 2020 were used to populate the connectivity matrices. Purple boxes show where more connections between countries were made in the daily run; green boxes show where more connections were made in the hourly run. Blank boxes show where no particles have connected between countries; boxes showing "0.0" have been rounded down but are in fact a non-zero value.

# Appendix C: Validation using drifters

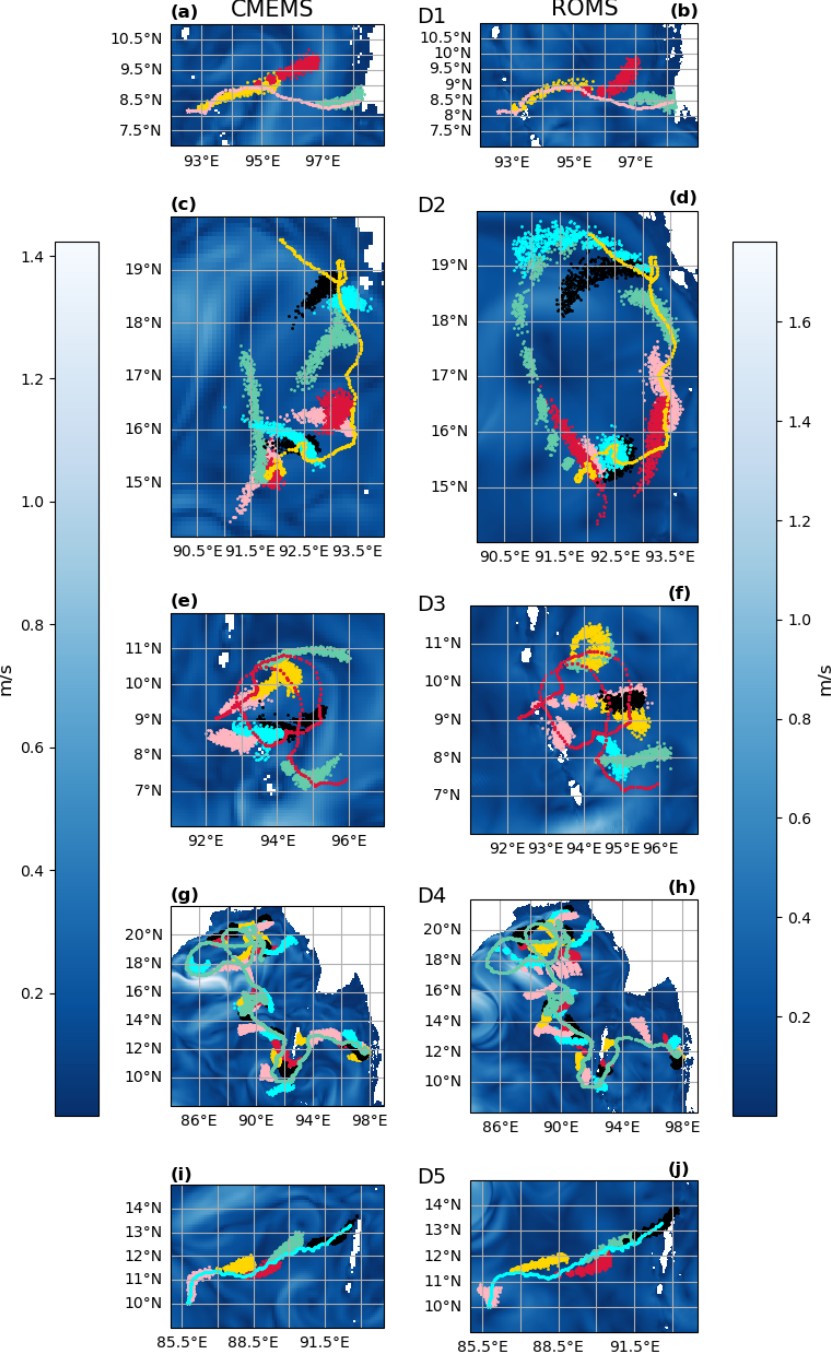

**Figure C1: Individual drifter tracks for each of the five drifters used for validation of particle tracking simulations of floating litter. Drifter tracks are colour-coded to correspond with those shown in Figure 2: D1 (pink); D2 (yellow); D3 (red); D4 (green); D5 (cyan). Particles are colour-coded by weekly release at the location of the drifter at that time and show how closely they follow the drifter tracks. See main text for details of experiments. Left panel shows CMEMS validation, right panel shows ROMS validation. Background is an arbitrary snapshot of ocean current speeds on 2nd June 2019 from each respective model.**

**Appendix D: Particle escape routes**

Monsoon (June – September)

| CMEMS | Exited domain through southeast boundary | | | Exited domain through southwest boundary | | | Exited domain through south boundary | | | Total |
|---|---|---|---|---|---|---|---|---|---|---|
| | Exited domain within first week | Exited domain during rest of season | Exited domain later | Exited domain within first week | Exited domain during rest of season | Exited domain later | Exited domain within first week | Exited domain during rest of season | Exited domain later | |
| Sri Lanka | 0% | 0% | 0% | 0% | 0% | 12% | 0% | 0% | 3% | 16% |
| India | 0% | 0% | 0% | <0.5% | <0.5% | 11% | 0% | 0% | 1% | 12% |
| Bangladesh | 0% | 0% | <0.5% | 0% | 0% | 1% | 0% | 0% | 0% | 1% |
| Myanmar | 0% | 0% | <0.5% | 0% | 0% | 1% | 0% | 0% | <0.5% | 1% |
| Thailand | 24% | 2% | <0.5% | 0% | 0% | 1% | 0% | 0% | 1% | 27% |
| Indonesia | 2% | 28% | 1% | 0% | 0% | 4% | 3% | 4% | 5% | 46% |

**Table D1: Percentage of particles released from each country in the monsoon season of the CMEMS simulation that exited the domain through each of the three open boundaries (see Figure 1) and how quickly. Particles that did not leave the domain in the season they were released may have been subjected to currents travelling in the opposite direction than expected.**

| ROMS | Exited domain through southeast boundary | | | Exited domain through southwest boundary | | | Exited domain through south boundary | | | Total |
|---|---|---|---|---|---|---|---|---|---|---|
| | Exited domain within first week | Exited domain during rest of season | Exited domain later | Exited domain within first week | Exited domain during rest of season | Exited domain later | Exited domain within first week | Exited domain during rest of season | Exited domain later | |
| Sri Lanka | 0% | <0.5% | 1% | 0% | 0% | 5% | 0% | 0% | 3% | 9% |
| India | 0% | 0% | <0.5% | <0.5% | 0% | 1% | 0% | 0% | <0.5% | 1% |
| Bangladesh | 0% | 0% | 0% | 0% | 0% | 0% | 0% | 0% | 0% | 0% |
| Myanmar | 0% | 0% | <0.5% | 0% | 0% | <0.5% | 0% | 0% | 0% | <0.5% |
| Thailand | 8% | <0.5% | <0.5% | 0% | 0% | <0.5% | 0% | 0% | <0.5% | 9% |
| Indonesia | 1% | 10% | 1% | 0% | 0% | 2% | 7% | 5% | 1% | 27% |

**Table D2: Percentage of particles released from each country in the monsoon season of the ROMS run that exited the domain**
**through each of the three open boundaries (see Figure 1) and how quickly. Particles that did not leave the domain in the season they were released may have been subjected to currents travelling in the opposite direction than expected.**

Post-monsoon (October – January)

| CMEMS | Exited domain through southeast boundary | | | Exited domain through southwest boundary | | | Exited domain through south boundary | | | Total |
|---|---|---|---|---|---|---|---|---|---|---|
| | Exited domain within first week | Exited domain during rest of season | Exited domain later | Exited domain within first week | Exited domain during rest of season | Exited domain later | Exited domain within first week | Exited domain during rest of season | Exited domain later | |
| Sri Lanka | 0% | 0% | 0% | 3% | 53% | 8% | <0.5% | 2% | <0.5% | 66% |
| India | 0% | 0% | 0% | 5% | 12% | <0.5% | 0% | <0.5% | 0% | 18% |
| Bangladesh | 0% | 0% | <0.5% | 0% | 1% | <0.5% | 0% | 0% | <0.5% | 1% |
| Myanmar | 0% | 0% | 1% | 0% | 1% | 6% | 0% | <0.5% | 5% | 14% |
| Thailand | 4% | 1% | 1% | 0% | <0.5% | 19% | 0% | 6% | 13% | 44% |
| Indonesia | 3% | 2% | 2% | 0% | <0.5% | 6% | 7% | 19% | 8% | 46% |

**Table D3: Percentage of particles released from each country in the post-monsoon season of the CMEMS simulation that exited the domain through each of the three open boundaries (see Figure 1) and how quickly. Particles that did not leave the domain in the season they were released may have been subjected to currents travelling in the opposite direction than expected.**

| ROMS | Exited domain through southeast boundary | | | Exited domain through southwest boundary | | | Exited domain through south boundary | | | Total |
|---|---|---|---|---|---|---|---|---|---|---|
| | Exited domain within first week | Exited domain during rest of season | Exited domain later | Exited domain within first week | Exited domain during rest of season | Exited domain later | Exited domain within first week | Exited domain during rest of season | Exited domain later | |
| Sri Lanka | 0% | 0% | 0% | 2% | 32% | 5% | <0.5% | 5% | <0.5% | 45% |
| India | 0% | 0% | <0.5% | 3% | 3% | <0.5% | 0% | <0.5% | <0.5% | 6% |
| Bangladesh | 0% | 0% | <0.5% | 0% | <0.5% | <0.5% | 0% | 0% | <0.5% | 1% |
| Myanmar | 0% | 0% | <0.5% | 0% | <0.5% | 4% | 0% | <0.5% | 1% | 5% |
| Thailand | 3% | 1% | 1% | 0% | <0.5% | 10% | 0% | 2% | 3% | 20% |
| Indonesia | 3% | 5% | <0.5% | 0% | 1% | 5% | 7% | 5% | 2% | 29% |

**Table D4: Percentage of particles released from each country in the post-monsoon season of the ROMS simulation that exited the domain through each of the three open boundaries (see Figure 1) and how quickly. Particles that did not leave the domain in the season they were released may have been subjected to currents travelling in the opposite direction than expected.**

Pre-monsoon (February – May)

| CMEMS | Exited domain through southeast boundary | | | Exited domain through southwest boundary | | | Exited domain through south boundary | | | Total |
|---|---|---|---|---|---|---|---|---|---|---|
| | Exited domain within first week | Exited domain during rest of season | Exited domain later | Exited domain within first week | Exited domain during rest of season | Exited domain later | Exited domain within first week | Exited domain during rest of season | Exited domain later | |
| Sri Lanka | 0% | 0% | <0.5% | 1% | 13% | 0% | <0.5% | 1% | 0% | 15% |
| India | 0% | 0% | 0% | 2% | 1% | 0% | 0% | <0.5% | 0% | 3% |
| Bangladesh | 0% | 0% | 0% | 0% | 0% | 0% | 0% | 0% | 0% | 0% |
| Myanmar | 0% | 1% | <0.5% | 0% | 0% | 0% | 0% | <0.5% | <0.5% | 1% |
| Thailand | 11% | 9% | 1% | 0% | 0% | 0% | 0% | 1% | <0.5% | 23% |
| Indonesia | 2% | 15% | 6% | 0% | 0% | 0% | 5% | 4% | 1% | 34% |

**Table D5: Percentage of particles released from each country in the pre-monsoon season of the CMEMS simulation that exited the domain through each of the three open boundaries (see Figure 1) and how quickly. Particles that did not leave the domain in this pre-monsoon season will have been subjected to monsoon currents later, which travel in the same direction.**

| ROMS | Exited domain through southeast boundary | | | Exited domain through southwest boundary | | | Exited domain through south boundary | | | Total |
|---|---|---|---|---|---|---|---|---|---|---|
| | Exited domain within first week | Exited domain during rest of season | Exited domain later | Exited domain within first week | Exited domain during rest of season | Exited domain later | Exited domain within first week | Exited domain during rest of season | Exited domain later | |
| Sri Lanka | 0% | 0% | <0.5% | 1% | 10% | 0% | <0.5% | <0.5% | 1% | 12% |
| India | 0% | 0% | 0% | 1% | <0.5% | 0% | 0% | <0.5% | 0% | 2% |
| Bangladesh | 0% | 0% | 0% | 0% | 0% | 0% | 0% | 0% | 0% | 0% |
| Myanmar | 0% | 1% | <0.5% | 0% | 0% | 0% | 0% | <0.5% | <0.5% | 1% |
| Thailand | 6% | 6% | 2% | 0% | <0.5% | 0% | 0% | <0.5% | <0.5% | 14% |
| Indonesia | 1% | 5% | 5% | 0% | <0.5% | 0% | 6% | 4% | 1% | 23% |

**Table D6: Percentage of particles released from each country in the pre-monsoon season of the ROMS simulation that exited the domain through each of the three open boundaries (see Figure 1) and how quickly. Particles that did not leave the domain in this pre-monsoon season will have been subjected to monsoon currents later, which travel in the same direction.**

**Code and data availability**

The code repository containing the model run scripts, data analysis code and scripts to generate the figures in this manuscript
is archived here: https://doi.org/10.5281/zenodo.15230045. All data created and used for analysis in this paper can be accessed
on the Cefas Data Portal: https://doi.org/10.14466/CefasDataHub.160. ROMS model simulated data presented in this paper
are archived at the central data repository of https://incois.gov.in/ and can be obtained by contacting kunal.c@incois.gov.in.
All other forcing data used in this study is publicly available from Copernicus through the Marine Data Store
(https://data.marine.copernicus.eu/products) or Climate Data Store (https://cds.climate.copernicus.eu/). Observed drifter
trajectories for validation were obtained from the Global Drifter Program: https://doi.org/10.25921/7ntx-z961.

**Author contribution**

Conceptualization: Lianne Harrison, Jennifer A. Graham; Data curation: Lianne Harrison; Methodology: all authors; Formal
analysis: Lianne Harrison; Writing - original draft: Lianne Harrison; Writing - review & editing: all authors

**Competing interests**

The authors declare that they have no conflict of interest.

**Acknowledgements**

The research presented in this paper was carried out on the High Performance Computing Cluster supported by the Research
and Specialist Computing Support service at the University of East Anglia. This work has been carried out under the
Memorandum of Understanding signed between the Ministry of Earth Sciences, Govt. of India and the Centre for Environment,
Fisheries and Aquaculture Science (CEFAS), UK for marine litter and micro plastics research. The authors would like to thank
and acknowledge the [Defra on behalf of the] UK government for funding this work: project number GB-GOV-7-BPFOCPP.
This study has been conducted using E.U. Copernicus Marine Service Information: 10.48670/moi-00016; 10.48670/moi-
00017. This is INCOIS contribution number xxx. The authors wish to thank Richard Heal for his helpful contributions when
designing the model and David Haverson and two anonymous reviewers for their comments which have greatly improved the
manuscript.

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
