# Peer review of "Monsoonal influence on floating marine litter pathways in the Bay of Bengal"

_EGUsphere, 2024_

## Author Comment (AC1)

Review 1

**Review to Monsoonal influence on floating marine litter pathways in the Bay of Bengal**

**General comments**

This manuscript deals with the connectivity of floating macro-plastics released along the coast in the Gulf of Bengal. This region seems to be an under-sampled and under-studied area from a Lagrangian point of view. In this way, the results of this manuscript are of interest. The manuscript is clear and well written. The approach of initializing Lagrangian scenarios uniformly along the coasts to overcome the major uncertainties in source estimates is interesting. However, there are some limitations and important methodological points that need to be addressed to finalize the study before it can be published.

From the plastic problem point of view, I regret the country-by-country approach of this study, which leaves each country to its own responsibilities instead of promoting collaborative regional approach as it is recommended by scientists and NGOs in the context of current international negotiations for a treaty against plastic pollution. For example, countries upstream of watersheds share the responsibility for marine pollution, and measures must be discussed with continuity at borders to ensure fairness towards countries with long emitting and accumulating coasts (such as Myanmar designated in this study). A alternative in future work would be to segment the coasts according to ocean dynamics or sub-region land use or coastal types.

We recognise that reporting by country might overlook issues such as shared watersheds. It does also have advantages, namely making results directly comparable between studies and quantifying progress once measures are taken. Intentionally, the litter sources were all normalised in this study, rather than providing numbers of mass export between countries, so that the focus is on the efficiency of the marine pathways. This motivation has been added to the beginning of the discussion section (lines 306-309).

I suggest adding a limitations section to the methods and/or discussion. We understand that the authors use Ocean Parcels by activating the options already coded in the tool. However, the processes chosen and activated should be at least described and discussed, mentioning particularly their limitations. It is important to be able to interpret the results in the light of these uncertainties. The 2D approach, for example, that might over-estimate the beaching rates (no undertow for mass conservation as in 3D) should be discussed, as does the simple addition of Stokes drift to 2D Eulerian currents (no anti-Stokes force created contrary to coupled simulations).

A new subsection ("Model limitations"; lines 164-203) has been added to the Methods section, which substantially details the constraints of the model and datasets used. A note regarding the use of 2D surface velocities instead of a full 3D simulation has been included as follows (lines 201-203):

"The use of only surface velocities rather than running a 3D simulation further limits the movement of particles. However, despite the particle tracking simulations being limited to 2D, the hydrodynamic simulations were run in 3D and this mitigates some of these shortcomings."

Since the advection kernel that is inbuilt in Parcels is central to the model's operation, it is already discussed extensively in papers describing the model development. Instead of repeating what has been published elsewhere, we have referred the reader to the relevant

citation. The diffusion kernel, which is not discussed in these articles, has now been described and the GitHub repository which houses the code for the two kernels used in this study has also been given a DOI and this is cited here so the reader can explicitly see how the kernels operate (lines 95-104):

"Advection of particles via surface ocean currents (detailed below) was included using an inbuilt OceanParcels kernel which uses a fourth-order Runge-Kutta advection scheme (Advection RK4, described in Lange and van Sebille, (2017)). Stokes drift velocities were included to account for the movement of particles resulting from wave motions by simple addition to surface current velocities. To account for sub-grid scale processes, diffusion is implemented as a random walk, through an inbuilt kernel known as DiffusionUniformKh. A diffusion coefficient of 100 $m^2$/s was chosen based on grid cell size (Peliz et al., 2007), as detailed below. The diffusion kernel combined this coefficient with a random variate calculated from a normal distribution with a mean of zero and standard deviation equal to the square root of the model timestep (see https://doi.org/10.5281/zenodo.14906471 where a copy of the advection and diffusion kernels used have been archived)."

Moreover, the manuscript is based on comparison of stranding between two simulations with different resolutions. Even if 2 km is a high resolution for regional approach, neither of the two runs has sufficient resolution to represent coastal and beaching processes, the dynamics in the coastal grid cells will still be very different (as mentioned by the authors in the section 4.3 but unfortunately not showed with figures). I suggest that the authors need to further clarify the particle release strategy in the two simulation, depending on the size of the grid cells, which is decisive for the comparison of beaching statistics (a figure zooming on coastal cells of both simulations with the particle release locations would be welcome, as well as one showing the final locations when they are considered beached in both simulations). This would illustrate how the resolution numerically constrain the release scenarios and beaching statistics, in addition to the representation of the dynamics.

Figure 1 has been updated to show the particle release locations with respect to the different coastlines in the CMEMS and ROMS models. This figure also now demonstrates the differences in coastal dynamics that are resolved by the different models as the backgrounds have been changed to show an ocean velocity snapshot for each of the models. Subfigures have also been included to show a small area in the southeast of the domain that has been enlarged to illustrate the differences in coastal dynamics. Moreover, a new figure has been added to Appendix A to show the final locations of particles that were considered beached on each of the coastlines so the proximity of the particles to the coast upon beaching can be seen more clearly. Additionally, the newly added "Model limitations" subsection discusses the challenge of resolving beaching processes at these model resolutions (lines 178-185):

"Regardless of the beaching method employed, the resolution of all these hydrodynamic models, including the two used in this study, are too coarse to fully capture all processes that are key to marine debris beaching. Fine-scale ocean dynamics such as submesoscale and microscale eddies near the coast contribute to litter accreting and washing ashore but are not represented even in the finer scale ROMS model we used to advect particles. Sub-grid scale tidal motions at the shoreline are also precluded, yet they would likely lead to higher beaching rates (Zhang et al., 2020), and slope at the coastline is not represented by either model. Additionally, the shape of the coastline, while much more realistic in the ROMS model versus CMEMS (Fig. 1), is still not refined enough to show the true morphology of the coast and misses

many features such as estuaries which have been demonstrated to act as traps for floating debris (Duncan et al., 2020; Pawlowicz et al., 2019)."

**Detailed comments**

L33. "Despite the large uncertainties": this is very important in the current challenges of quantifying sources and monitoring plastic pollution in the marine environment, I suggest to add the orders of magnitude of these uncertainties here referring to the recent literature on the subject (interesting studies have followed the precursory but not up-to-date study by Jambeck et al., 2015).

While there are a large number of studies that have looked into this problem, we are only aware of one other that has reported a global estimate of plastic waste entering the ocean from (almost) all sources at the coast. Many others are not global studies and only focus on the sources of mismanaged waste to the ocean in small regions. A few only discuss pollution coming from rivers, and while this makes up the majority of input into the ocean, it does not encompass other litter sources as in the Jambeck et al. (2015) study. Additionally, our manuscript has taken the approach that river inputs to the ocean have been shown to not produce accurate source-to-sink estimates for floating marine litter. The most recent, similar estimate we can find (Borrelle et al., 2020) details waste finding its way to all aquatic environments, not solely the ocean. Therefore, order of magnitude error estimates based on studies that have looked at waste input to the ocean based on different criteria are not really possible to calculate. The estimate quoted from Jambeck et al. (2015) is one of two studies we are aware of that are relevant to our research and their estimate is purposely given as a range to indicate the uncertainty in the calculations.

We have instead emphasised the level of uncertainty by adding in details of other drastically different estimates to show how they can differ, but stress that they have estimated different things (lines 31-43).

"Jambeck et al. (2015) estimated that between 4.8 - 12.7 million tonnes of plastic entered our oceans every year (based on conditions in 2010), and that this could increase by an order of magnitude by 2025. Other studies have calculated significantly different estimates for how much plastic finds its way to the ocean. Lebreton and Andrady (2019) calculated a lower estimate of between 3.1 – 8.2 Mt of plastic entering the ocean each year, using a different dataset for solid waste generation but similar assumptions to that of Jambeck et al. (2015) about how much mismanaged waste within 50 km of the coast finds its way into the ocean. Several other studies have investigated slightly different questions and about how much plastic waste enters the oceans which have resulted in quite different estimates. Lebreton et al. (2017), Schmidt et al. (2018) and Meijer et al. (2021) calculated how much plastic is transported solely by rivers to the ocean, resulting in much lower values of 1.1 - 2.4 Mt/yr, 0.5 – 2.8 Mt/yr, and 0.8 – 2.7 Mt/yr, respectively. A subsequent study by Borrelle et al. (2020) found a significantly larger value of 19 – 23 Mt of plastic waste ending up in aquatic environments in 2016, however, this includes rivers and lakes rather than just the ocean. Despite the large uncertainties associated with these estimates, observations of so-called 'garbage patches' that have formed in the ocean's major gyres (Cózar et al., 2014; Eriksen et al., 2014) and reports of litter washing up on beaches (e.g. Shankar et al., 2023) confirm plastic pollution in the ocean is a vast problem."

L41. "with fewer looking at the connections of litter that 'beaches', or washes ashore, along coastlines": seems exaggerated, many studies have indeed focus on the gyres, well represented

by the large-scale dynamics, whereas coastal processes are more complex but there is still a significant literature on Lagrangian tracking applied to plastic beaching issue. An updated state of the art would be welcome.

We have removed this text.

L42. "two-thirds […] is captured on coastlines": I suggest to note the huge uncertainties in the statistics mentioned here given the resolution of the global models cited.

We have altered to wording of this sentence to make it clearer that these estimates are approximate and have explicitly noted the large uncertainties associated with them (lines 48-50):

"However, multiple recent studies have suggested that approximately two-thirds to three-quarters of all litter released in global model simulations may be captured on coastlines, though they note there are large uncertainties associated with these estimates (Chassignet et al., 2021; Chenillat et al., 2021; Lebreton et al., 2019; Onink et al., 2021)."

L59. "Lebreton et al. (2017), which have very high uncertainties": others river input models have shown the sources of uncertainties in the mentioned reference, add citations

Citations which discuss the large uncertainties and some of the sources of uncertainty have now been included here.

L93. "Stokes drift […] wave motions": How is added the effect of Stokes Drift on the 2D current fields? A simple addition is physically very different from a coupling process for example

We have clarified that the velocities were combined through simple addition (lines 97-98):

"Stokes drift velocities were included to account for the movement of particles resulting from wave motions by simple addition to surface current velocities."

L95. "Windage […] trajectories": for which macro-plastic size are these 1% consistent ? Is there a risk of obtaining excessively high drift velocities for particles by combining all these effects?

In response to this and a similar comment by another reviewer, the sentence referred to here has now been altered and this section has been updated to give more details of how we came to the decision to use a 1% windage coefficient in addition to Stokes and surface ocean velocities. This is an approach used in other studies (e.g. Chassignet et al., 2021; Isobe and Iwasaki, 2022) and is based on the physical processes that need to be considered for the movement of floating litter (Haza et al., 2019). While the effect of wind on the surface currents themselves is already included in the ocean current velocities used, the addition of windage takes into account the extra push that wind provides as a result of friction against the portion of marine debris that is not under the surface of the water. This is the reason the additional wind factor is necessary without 'double-counting' the wind's effect that one might suspect would lead to excessively high drift velocities. This explanation has been added to the text (lines 104-108):

"Windage is implemented in the model by applying 1% of the wind velocity to the particles' trajectories. Following analysis of observations of the wind's effect on undrogued drifters by Pereiro et al. (2018), this should describe all but very buoyant items of litter. While the effect of wind on the surface ocean currents is already included in the ocean velocities used in the particle tracking simulations, the addition of windage takes into account the extra push that

wind provides as a result of friction against the portion of marine debris that extends above the surface."

L114. "Particles [...] locations.": did the particles have exactly the same release lon,lat in both simulations given the different resolutions of coastal cells and the same distance to the land mask?

The particles did have exactly the same lat/lon release locations in both simulations, but as the shorelines were in slightly different locations in each model due to the resolution, the distance between the particles and the shoreline was different for particles in different locations in each model (i.e. some particles are further away from the coastline in the ROMS model than they are in the CMEMS model whereas some are closer, and vice versa). We have clarified that the mean and maximum distances stated were from the Natural Earth coastline and stated explicitly that the release locations were exactly the same in both simulations and the differences can now be seen in a figure (lines 133-140):

"Particle release locations were uniformly spaced around all major coastlines in the Bay of Bengal (Fig. 1). Particles were released on average 6 km from the Natural Earth coastline (naturalearthdata.com), with a maximum distance of 18 km in some locations. This distance was chosen to complement different coastlines from the two hydrodynamic models, ensuring no particles were released on land while also ensuring they were released on the continental shelf for both configurations; this ensured coastal dynamics rather than open ocean dynamics influenced the particles when they initiated their journeys. We chose to release the particles from exactly the same latitudes and longitudes in both simulations, but note this means their proximity to the coast will differ between the CMEMS and ROMS runs, due to the differences in hydrodynamic model resolution (Fig. 1c-d)."

L122. "we could be applied [...] future": this perspective may be little excessive, the number of particles released per day seems low and the question of statistical representativeness of the diversity of possible trajectories is not addressed ("500 coastal locations every day, [...] with 182,500 particles released in total"). The authors should add a statistical sensitivity test to show that the number of particles released is sufficient to represent the diversity of particle fates in the studied region (i.e. increase the number of particle releases and see if it changes or not the connectivity statistics, taking into account the spatial and temporal variability of the dynamics - in the same way they have done the temporal resolution sensitivity analysis described in Appendix A).

We acknowledge the importance of testing the sensitivity of the results to the number and distribution of released particles. While we have not conducted an additional sensitivity test, the consistency of the patterns observed across both simulations suggests that the conclusions are robust to variations in particle release parameters. In addition, we ensured that the particle release strategy followed established practices in similar studies to achieve statistically representative connectivity patterns. Our approach to the number of particles released per day is in line with, or in excess of, all other studies of this nature cited in the manuscript. van der Mheen et al. (2020a) released a similar number of particles to our study (~200,000 in total over the course of a year) in the northern Indian Ocean, which was a larger geographical area that included the Arabian Sea, adjacent to the Bay of Bengal. Irfan et al. (2024) released far less than this in their study of the Bay of Bengal because they chose to release particles monthly. This low frequency results in a total number of particles that is an almost 10 times lower than our study over their 10 year run (~200,000). Many global studies

have used a range of larger numbers of total particles released per year but considering the geographical coverage is so much larger, our statistical robustness is greater. Examples include: Lebreton et al. (2012): ~120,000 - 500,000 per year; Chassignet et al. (2021): ~350,000; Chennilat et al. (2021): ~240,000; Onink et al. (2021): ~600,000 per year. Many of these global simulations have also used monthly releases as opposed to our more frequent daily releases. It should also be noted that many studies have released different numbers of particles depending on the assumed amount of litter entering the ocean from a given location and this changes throughout the year (e.g. Chassignet et al., 2021) meaning that in some locations, a single particle is released per month to represent a small amount of litter.

The manuscript has been updated to give some of these examples which has set the precedent for our method (lines 140-143):

"A particle was released from each of the 500 coastal locations every day for a year, with 182,500 particles released in total. The number of released particles is consistent with other particle tracking studies conducted in the Bay of Bengal and on a global scale (e.g. Chassignet et al., 2021; Chenillat et al., 2021; Lebreton et al., 2012; van der Mheen et al., 2020a)."

L150. Are the 100 particles really released at exactly the same position and at exactly the same time? So why do the 100 differ from one another?

Diffusion is implemented as a random walk at each time step which leads to the differences. We have clarified this in the manuscript (line 222):

"Note that the random-walk diffusion causes each of the 100 particles to take a slightly different path."

Figure 2. It would be interesting to be able to see the weekly particle clouds (like Fig. 2b) for each D1-4 drifter trajectory in the supplementary materials, to see the spread associated with the statistics in Fig. 2c (for each of the two simulations). Also, does Fig. 2b correspond to CMEMS or ROMS advection? Please specify.

A figure has been added to Appendix C to show the validation comparisons for each model and each drifter. The caption for Figure 2 has been altered to identify that the images in Fig. 2a&b are from the ROMS run but that the CMEMS figures and the rest of the drifters for both runs can be seen in Appendix C.

L192. It would be welcome here to have, for example, a current map to understand why the exit patterns are different, and based on the bibliography of dynamics in the region, which seasonal pattern is predominant over the years. I did not find the corresponding circulation analysis in the Discussion section.

The different currents can best be seen through the supplementary animations. The particles can be seen leaving the domain as the seasons progress. Readers have now been pointed towards these animations when outlining the differences in CMEMS and ROMS simulations. Additionally, we have included extra details in the post-monsoon section of the Results that this season dominated the year-long simulation in terms of total escaped particles (lines 271-277):

"The number of remaining particles leaving the domain was also higher than the monsoon and pre-monsoon seasons combined (CMEMS: 27%, ROMS: 15%) by a substantial margin, making the particle-exit pattern from this post-monsoon season dominant across the full simulation for the CMEMS and ROMS cases. The majority of these particles left the domain through the

southwestern boundary towards the Arabian Sea (CMEMS: 19%, ROMS: 10%), a smaller portion leaving through the southern open boundary (CMEMS: 7%, ROMS: 3%), and relatively few leaving though the southeastern boundary into the Strait of Malacca (1%) in each case. These groupings are reflected in the particle-exit pattern for the full simulation in both cases."

L243. The fact that beaching is predominant in the vicinity of source points in all literature studies at these modeled resolution does not confer an element of validation since none of these studies allows the representation of realistic beaching. I would advise more nuance with regard to the numerical limitations in these assertions, especially as the study's beaching criterion is rather simplistic.

We have added a caveat to this paragraph to address the limitations of modelling studies on this scale to effectively simulate beaching processes (lines 312-315):

"This is consistent with previous modelling studies in the region (e.g. Chassignet et al., 2021; Chenillat et al., 2021) and is unsurprising given that the resolution of global or regional-scale models is insufficient at the coast to implement realistic beaching processes and instead, simpler beaching methods are employed in this study and others."

L267. The quantification of connectivity between countries seems totally linked to particle emission scenarios, the argument brought by the authors at the end of the section is in fact central to the differences observed compared to Chassignet et al., 2021 and should be mentioned right at the beginning of the paragraph: the sink differences should be discussed in relation to the differences in sources of the cited publication (in term of quantity and location).

This paragraph has been moved to the beginning of the discussion about comparisons to the results of Chassignet et al. (2021) so that readers can keep this in mind when we discuss the similarities and differences of our findings.

L282. I was not able to see the Supplementary animations. I suggest that this section 4.2 could be illustrated by one or two figures of ocean circulation to help understand the different seasonal pathways discussed and the importance of simulation resolution for the study of connectivity.

Figure 1 has been updated to reflect other comments by yourself and the other reviewer and now shows the differences in resolution between the CMEMS and ROMS simulations (e.g. smaller eddies are resolved in the ROMS simulation). Animations are available alongside the preprint under Assets > Supplement. These animations show seasonal pathways far more clearly than a set of static figures would as the journeys of the particles clearly show the pathways of currents at different times of the year.

L335 - 339. The assumptions made here should be illustrated by coastal zoom circulation maps (eddies and/or offshore current) at the two simulation resolutions used. Otherwise remove as unfounded.

Figure 1 has been updated to show a snapshot from each model for the whole domain and a zoomed in area around Indonesia and Thailand which demonstrates that the higher resolution ROMS velocities capture smaller scale features that could not be resolved in the CMEMS model. An example of coastal currents eddying off the northern tip of Indonesia are depicted in the snapshot chosen. The comparison shows that these features are present but not as well resolved in the CMEMS data. The text referred to here has been updated to point towards this new figure (lines 428-429):

"An example of the difference in the level of detail of coastal currents and mesoscale eddies resolved by each of the models can be seen in Fig. 1c-d."

L341. Has the CMEMS product assimilated the drifters' profiles? If so, it's normal that the CMEMS simulation corresponds better to the observations even if its resolution is coarser.

No, we do not believe the drifter velocities have been assimilated. The CMEMS product has only assimilated data related to sea level, sea ice concentration and/or thickness, SST, and data from in-situ TS profiles from the sources listed below. It is possible that other observations from the drifters may have been included in some of the products listed here, but velocities would not have been assimilated based on the information provided in the product manual:

| Assimilated observations | L3S SST (ODYSSEA), SIC (OSI SAF), SLA (AVISO), T/S profiles (CORIOLIS database) MDT adjusted based on CNES-CLS18, Mulet et al., 2021 WOA 2013 climatology (temperature and salinity) below 2000 m (assimilation using a non-Gaussian error at depth) |
|---|---|

We have updated the information in the manuscript to include this detail (in the Methods, lines 213-215):

"This is an important consideration given that the CMEMS simulations include data assimilation (for sea level, temperature and salinity) and would therefore be expected to provide more accurate offshore currents than the ROMS velocities."

L360. Even with Stokes drift and windage included in the study, coastal processes are not represented: put more nuance in this sentence. Moreover, Stokes drift seems to be added by simple addition with the Eulerian current fields: here again, this is a strong limitation to be discussed. In coupled simulations, Stokes drift forcing creates a feedback from the current fields called anti-Stokes force that attenuates the total current compared to the total current obtained by adding the Eulerian current and Stokes drift without coupling at each time step.

The limitations set by the simple addition of Stokes drift velocities to those of the Eulerian currents has been addressed in the limitations section added to the methods, as detailed above. We have altered the wording in this sentence to make it clearer that wind and waves are not the only drivers of beaching, just contributors. Additionally, we have added a caveat to the end of the paragraph to state that coastal processes responsible for beaching are not resolved in the models used in this work (lines 453-460):

"Their model did not feature key mechanisms thought to promote the beaching of floating particles, such as windage or Stokes drift, instead assuming a beaching probability. Winds and waves are likely to have a large effect on beaching probabilities; Stokes drift, for example, has been found to reduce the residence time of particles in simulations in the Black Sea as well as increasing beaching rates by up to 75% (Castro-Rosero et al., 2023). Onink et al. (2021) found that not including Stokes drift in their global model reduced the trapping of particles near the coast and reduced beaching by 6-7%. Additionally, Irfan et al. (2024) found increases in beaching rates of 5% when Stokes drift was added to their model and a further 9% when windage was included. It is important to note, however, that neither model used in this study is able to fully resolve all the coastal processes that are likely to influence beaching rates."

L380-383 Same remark as in the method section. To ensure that the conclusions drawn from the study's Lagrangian simulations can be extended to different cases by weighting the particles according to various source scenarios, I suggest adding a sensitivity test in the appendix showing that the number of particles released is sufficient for the connectivity obtained to be statistically representative: a comparison of the spread and connectivity of clouds composed of different numbers of particles with slightly different time and space initialization should be added (with a histogram as in Fig. 2c for example).

In our simulations, we ensured that the particle release strategy followed established practices in similar studies (van der Mheen et al., 2020a; Irfan et al., 2024; Lebreton et al., 2012; Chassignet et al., 2021; Chennilat et al., 2021; Onink et al., 2021), as detailed above, to achieve statistically representative connectivity patterns. While we have not conducted an additional sensitivity test, the consistency of the patterns observed across both simulations suggests that the conclusions are robust to variations in particle release parameters.

Following similar comments from another reviewer about the likelihood of our results being weighted with updated pollution estimates, we have removed the text here emphasising the future applications of the model as a tool that could be reused with newer, more accurate weightings applied to our results. Therefore, while we agree this sensitivity analysis would be very interesting to investigate, this piece of work would be a very large undertaking that is outside the scope of this paper, especially given the lack of emphasis now given to the future weightings, and would not add enough value to the manuscript to justify the work.

L400. "Our simulations [...] litter." This recommendation is far too vague and should be removed: targeting the entire Myanmar coast is illusory, and the study offers no way of targeting smaller coastal transect suitable for the scale of a beach litter observation.

We have removed this sentence.

L404-405. Same comment as above, the present connectivity study remains interesting, but the scale of the coastline considered here is not suitable as support for policy decisions and beach cleaning operations (which requires a link to finer scales). I suggest nuancing the last two sentences or removing them.

The point about needing finer scale research to influence policy decisions and targets for beach cleaning is valid. Following other comments from yourself and another reviewer, this paragraph has been removed from the conclusions section.

Appendix A. L433. "The differences in sink locations [...] negligible": I would put more nuance into this kind of statement, the connectivity presented here is calculated between very long lengths of coastline, if smaller transects had been taken into account for the connectivity study, the differences might be higher.

We have altered this statement to make clear that the coarse resolution of the polygons used for connectivity calculations has contributed to our conclusion that the differences between the hourly and daily results were negligible (lines 520-523):

"Considering the aim of this study was to quantify connections between just six countries in such a large domain, as opposed to a more granular breakdown of the region, the differences in sink locations between the hourly and daily runs for each case (CMEMS and ROMS) were found to be negligible. Therefore, daily forcing was determined to be adequate to provide accurate results for the final experiment."

Figure A1. Rather than giving the difference between the two connectivity matrices for each simulation, give the percentage error

The difference between the numbers of particles that were released from and beached on a given country in each simulation (hourly versus daily) is given in this figure. This provides an estimate of the margin of error that occurs in the separate simulations. As the values are given as a percentage of the number of particles released from each country, we would argue this figure already shows the percentage error. As this format remains consistent with the other connectivity matrices in the manuscript, we have opted to keep this figure in its original format.

**References**

[revised manuscript text omitted]

---

## Author Comment (AC2)

Review 2

**Main comments**

This study concerns the dynamics of floating plastic waste in the Bay of Bengal based on particle tracking simulations. Using a uniform release of particles along coastlines in the Bay of Bengal, the authors find that most particles beach in their country of origin and that the seasonally reversing East India Coastal Current (EICC) is the main driver of particle transport in the region. The study is relevant, and the manuscript is well-written and has high-quality figures. I have two main comments, which I believe can be addressed with additions and clarifications to the manuscript text and figures, with perhaps some additional analyses.

My first main comment concerns the approach to beaching in the study and, as an extension of that, the release locations. Since the connectivity between countries in the Bay of Bengal is determined based on beached particles, I think the assumptions and limitations of the beaching method should be thoroughly discussed in the manuscript. The authors mention in the Discussion that their "model accounts for several processes ... key mechanisms thought to drive the beaching of floating particles, such as windage or Stokes drift". They also discuss that the higher resolution ROMS model likely incorporates coastal dynamics more accurately than the global CMEMS model. While I agree with both these points, I think it is important to note that:

1.  Although the ROMS model may have a higher resolution and may capture coastal dynamics more accurately than the global CMEMS, both the Stokes drift and wind fields are based on global models and have coarse resolution (I assume ~25 km, although this is not specified). Since these are the mechanisms that are responsible for 'beaching' in the simulations, I think it is important to clarify that these are unlikely to represent fine-scale coastal dynamics and therefore are probably not capturing accurate beaching dynamics of floating particles.

    Additional details about the resolution of the wind and Stokes datasets were missing from the manuscript – thank you for pointing this out – and have now been included in the Methods (lines 128-132):

    "Stokes drift velocities were available in 3-hourly timesteps at a resolution of 1/5°, which is roughly 21 km at the latitudes of the Bay of Bengal; wind velocities were hourly and with a resolution of 1/4°, which is approximately 26 km at these latitudes. Both datasets were interpolated onto the relevant grid for each of the CMEMS and ROMS runs using cubic interpolation and then averaged to daily timesteps."

    Further discussion about the use of these datasets in both simulations despite their coarser resolution compared with the ocean velocity datasets has now been added to the Model limitations subsection within the Methods description (lines 194-197):

    "While the ocean current velocity datasets, particularly the ROMS data, have relatively high spatial resolution for a regional model such as this, the Stokes drift and wind velocities used in both the CMEMS and ROMS simulations are coarser than the ocean velocities. Any differences in beaching between the two simulations is therefore expected to result from the differences in general circulation patterns as opposed to wind and wave effects."

2. The beaching of particles is not simulated explicitly in this study, instead particles simply become stuck on land if they end up on a coastline (approximated in the study by identifying near-zero velocities rather than using a land mask). Since there is currently no consensus on the correct way to simulate beaching, I do not necessarily have an issue with this method, but I think the authors should make this clear in the manuscript. There are several studies that use different (probabilistic) approaches to beaching (e.g. Onink et al. 2021; Irfan et al. 2024; van der Mheen et al. 2020b; all already referenced in the manuscript); it would be good to briefly mention the different approach in these studies and explicitly clarify the approach used in this study in the Methods.

Different approaches taken by other researchers have now been mentioned in the Model limitations subsection and the method used by others which is most similar to the method we employed (which is stated explicitly earlier in the Methods section) is pointed out (lines 169-177):

"The beaching process is a critical step in the journey of a piece of marine litter (Hinata et al., 2020a) yet there is no consensus on how best to implement this step in particle tracking models. Some researchers have used a similar method to this study whereby a particle was deemed to be beached when its position was on a land grid cell (e.g. Irfan et al., 2024), whereas others considered particles beached if they persisted in a coastal grid cell for a given amount of time (e.g. Isobe and Iwasaki, 2022). Several other studies have taken the approach of probabilistic determination. For example, van der Mheen et al. (2020a) used a random probability to determine if a particle would beach, so long as it was within a given distance of the coast and that distance was decreasing. Chenillat et al. (2021) and Onink et al. (2021) used similar methods to this. Nevertheless, each of these methods used to determine particle beaching are simplistic and neglect much of the nuance involved with beaching processes in reality. Therefore, this study acknowledges the limitations of using this approach to determine particle beaching."

3. I think it is also important for the authors to clarify whether it is possible for particles to become stuck on land because of ocean currents (which is sometimes the case when ocean currents are interpolated incorrectly along coastlines but is not physically valid) or because of the random motion of particles, or if they implemented some method to only allow particles to end up on land as a result of Stokes drift and windage (which, as the authors also mention in the Discussion, would be the only physical mechanism to result in beaching). If no method was implemented to ensure this in the simulations, it would be useful if the authors could provide an indication of how many particles may become stuck on land (in areas on non-zero velocity) due to model artifacts (e.g. ocean current interpolation and random motion) versus due to Stokes drift and windage, though I appreciate this may be difficult to determine. In addition, the authors use the condition that a particle's velocity is close to zero to determine if a particle has beached. I assume that wind velocities have been set to zero above land? Since the land boundary is effectively identified by zero velocities, does this mean that "land" (or the area of zero velocities) is formed by a combination of the different velocity fields at different resolutions (rather than, for example, taking the land mask from just the ocean current models)? It would be useful to briefly explain this in the methods, and if "land" is indeed identified as a combination of all velocity fields, I think it would be useful to show maps (perhaps in an Appendix) of what this looks like for both the CMEMS and the ROMS simulations.

The inclusion of horizontal diffusivity representing subscale turbulence will result in beaching even if the currents are perfectly interpolated. In the model results it is not possible to discern when beaching resulted for this or from Stokes drift or windage aimed at the shore, but in reality it is not possible to relate this with observations.

Wind over land has not been set to zero as wind does not stop at land boundaries. So, it is true that windage may still contribute to particle motion over land. However, "land" is determined in the model only by near-zero ocean velocities. This was already stated in the manuscript. However, based on the reviewer's comments, we have added further clarification as follows (lines 109-112):

"At the end of each timestep, after advancing each particle's position, ocean velocities were checked at this new position. If the ocean velocity was less than $10^{-14}$ m/s, the particle was considered to be beached (after Delandmeter and van Sebille (2019)) and was no longer tracked. Stokes drift and wind velocities were not included in the calculations to determine if a particle was beached."

4. Regarding the release locations: I think using a uniform release along coastlines is an excellent method to gain insights into the dynamics of floating plastic in the region, especially (as the authors also highlight) given the large uncertainties surrounding estimates of riverine and coastal plastic sources. However, I do think that the choice in release location (distance from the coast) may have an effect on the simulation results, especially since the emphasis is on the beaching of particles. I think the motivation of releasing particles away from the coast (to not release any on land) but still in the continental shelf region (to hopefully capture some coastal dynamics, rather than only open ocean dynamics) is the correct one. However, in the Methods it is mentioned that "this distance was chosen to complement different coastlines from the two hydrodynamic models". Considering that the coastlines from the hydrodynamic models are not actually used as "land" to determine beaching, I am not sure if this is the correct motivation. I do not necessarily think that this needs changing in the simulations, but I think the wording here gives the impression that the ocean current models determine where beaching will occur, which (if I understood correctly) is not the case since the Stokes drift and wind field velocities also contribute to this. I would recommend showing maps with both the release locations and the zero-velocity "land" that is determined with the combined velocity fields (assuming that the land boundary shown in Figure 1 is a general boundary and not based on the zero-velocity region). Releasing close to the coast and considering the high beaching percentages on countries of origin also raises the question how many particles end up on land shortly after their release (see also my second point for this).

We have altered all figures to show the zero-ocean velocity boundary in each case, rather than the general boundary/coastline (as specified by Cartopy using Python) that was in the previous iteration of the maps. Figure 1 now shows the release locations near to the zero-velocity "land" for both the CMEMS and ROMS simulations.

Only zero-velocities in the CMEMS or ROMS ocean velocity datasets determined whether a particle was considered beached or not. As noted in response to the previous comment, we have altered the text regarding the determination of whether particles were considered beached, in case it was unclear.

We have also included new analysis and figures regarding how quickly particles beach following their release. Details are provided in answer to your second main comment below.

In addition to clarifying the points above in the manuscript, I think at least a paragraph in the Discussion should be dedicated to discussing the limitations of the simulation methods, with a particular focus on the beaching method. I think a discussion of the dependence on beaching on fine-scale local dynamics (which I think are unlikely to be captured in these simulations, despite the higher resolution ROMS model) should also be included. Some potential references for consideration for this are, for example: Pawlowicz et al. (2019), https://doi.org/10.1016/j.ecss.2019.106246; Zhang et al. (2020), https://doi.org/10.1016/j.scitotenv.2020.136634; Hinata et al. (2020a), https://doi.org/10.1016/j.marpolbul.2020.110910; and Hinata et al. (2020b), https://doi.org/10.1016/j.marpolbul.2020.111548.

A "Model limitations" subsection has been added to the Methods (following a similar comment from another reviewer), so that readers can keep these in mind when assessing the results. The limitations of the beaching process in the model were central to this discussion and all of these references have contributed. Thank you for the suggestions.

My second main comment concerns the discussion of the seasonal variations results (section 3.3). Figures 3c-h show connectivity matrices for particles released during the monsoon, post-monsoon, and pre-monsoon but beaching at any time during the simulation. Figure B1 shows the same connectivity matrices but for beaching only occurring during the specified monsoon season. I am not sure that Figure B1 should be in an appendix. I think showing both the connectivity matrices for beaching throughout the simulation and for beaching during the relevant monsoon season is important. For example, in the results for the monsoon season the authors identify that "the second highest beaching rate was always on a country in the anticlockwise direction" in Figure 3c, which does not make sense to me during the monsoon season, since the currents are in a clockwise direction. In Figure B1a this pattern doesn't seem to be as pronounced (e.g. second-highest beaching of particles from India is in Bangladesh rather than in Sri Lanka, which seems to make more sense with the direction of the ocean currents during the monsoon season). The connectivity matrices during the post-monsoon season in Figure 3 and Figure B1 also seem quite different. I think it is important to discuss these results along-side each other, as they can also provide information about how many particles tend to beach within the same season versus during a following season. Similarly, the mention of the different boundaries through which particles exit the region during the different seasons is interesting, but it does raise the question during which season these particles exited the region (not just in which season they were released).

In addition to showing the connectivity matrices, I think it would also be very relevant to show timeseries of the percentage of particles beaching and exiting the region (e.g. different panels per season release and different colours per country release). This would provide insights into how many particles beach very quickly after their release, within the same season of release, and in a different season.

A series of tables have also been added to Appendix D
Figure B1 and the related text has been moved into the results section of the main paper. Two additional figures have been added to the paper showing what percentage of particles released during each season beached or left the domain very quickly (within a week), later in the season of release, or later (in a different season). A series of tables have also been added to Appendix D

breaking down the numbers of particles exiting the domain via each domain boundary. These figures and tables have been referred to in several locations within the Results and Discussion and additional analysis has been included particularly in Sections 4.2 & 4.3.

**Minor comments**

I appreciate the validation of the ocean models with the undrogued drifter trajectories. However, I think it is important to make it clear from the start that these drifters provide validation in the open ocean only and not in the coastal ocean (where you may expect the higher resolution ROMS model to perform better than CMEMS). The authors make this clear in the Discussion, but I would recommend also briefly pointing this out in the Methods and Results sections about the validation. I would also be careful about including this in the Abstract, I personally think the sentence "Both simulations were validated using the pathways of undrogued surface drifters, with better agreement found for particles advected by data-assimilated ocean velocity" misses some nuance and may be misinterpreted.

The Methods have been updated to include information about the drifters mostly verifying open ocean dynamics to make the reader aware of this when they consider the results (lines 212-215):

"Most of these drifters began and continued their journeys in the open ocean. Consequently, minimal proportions of these drifter trajectories can be considered to verify coastal dynamics. This is an important consideration given that the CMEMS simulations include data assimilation (for sea level, temperature and salinity) and would therefore be expected to provide more accurate offshore currents than the ROMS velocities."

As this information is now stated in the Methods and Discussion, we have not repeated it in the Results which we wish to keep to the statistics; these results are discussed later on with this information in mind.

The sentence in the abstract has been updated to add some nuance (lines 22-24):

"Both simulations were validated using the pathways of undrogued surface drifters, which moved primarily within the open ocean, with better agreement found here for particles advected by data-assimilated ocean velocities."

The terminology around the monsoon seasons is a bit inconsistent throughout the manuscript (e.g. use of spring, summer, winter in the Introduction; pre-monsoon, monsoon, and post-monsoon in Methods, Results, and Discussion; Northeast Monsoon, Southwest Monsoon, Winter Monsoon in Discussion). I would recommend defining the pre-monsoon, monsoon, and post-monsoon seasons in the Introduction (the monthly definitions are now only given in the Methods). Since there is some different terminology used around the monsoon seasons in the region, it would also be beneficial to include the specific months you are referring to in all Figures and the Table.

The choice to refer to the same season by different names was a consequence of choosing to use the language utilised by other researchers in their respective papers, but we understand this may be confusing to readers, especially those unfamiliar with the monsoon and its seasonal changes. Therefore, we have removed almost all references to either the Summer/Winter Monsoon or the Southwest/Northeast Monsoon and instead refer to the relevant months to be clear about the time of year that is being discussed (e.g. Line 401: "This is in line with van der Mheen et al. (2020a) who found particles in their own simulations of the

wider northern Indian Ocean were transported from the Bay of Bengal into the Arabian Sea during December – February."). The only exception is line 58 where we refer to a season by another researcher's definition of it and this has been stated explicitly (lines 58-59):

"They concluded that beaching in the Bay of Bengal peaked on the north-northeast coastlines during the Southwest Monsoon (which they defined as June - October) but did not quantify beaching rates for each country."

I would appreciate some discussion about the choice of using ocean surface currents + Stokes drift + 1% windage to represent the transport of floating plastic waste. It is mentioned in the Methods that 1% windage best represents the effect of wind on drifters (undrogued?) but is there evidence that these drifters behave as floating plastic would? I do not have an issue with the choice of forcing fields, but I do think the choice should be discussed more. For example, while this may represent the transport of many very buoyant plastics (all but the largest, as mentioned in Methods), this may not correctly represent less buoyant or smaller plastics drifting at, or close to the ocean surface. I was hoping to see a sensitivity analysis on different forcing mechanisms, at least with and without windage added. It would be very interesting to see how this affects beaching in the simulations, but I appreciate this may be out of scope for the current study.

The type of sensitivity analysis you mentioned has already been done elsewhere (Irfan et al., 2024), so we do not wish to repeat that here. The methods have been updated to give more details of how we came to the decision to use a combination of ocean velocities + windage + Stokes drift. This is an approach used in other studies (e.g. Chassignet et al., 2021; Isobe and Iwasaki, 2022) and is based on the physical processes that need to be considered for the movement of floating litter (Haza et al., 2019) (lines 92-95):

"The model includes several processes which are believed to be the main physical processes responsible for influencing the movement of floating particles around the domain to simulate the dispersal of buoyant marine debris (Haza et al., 2019). This approach follows similar methods of others to simulate marine plastics distribution (e.g. Chassignet et al., 2021; Isobe and Iwasaki, 2022)."

Plastics of different buoyancies have been found to affect drift behaviour (Pereiro et al. 2018) but undrogued drifters are the closest analogy to floating litter that we can track to validate the model. A new study has used actual plastic bottles fitted with tracking technology (Duncan et al., 2020) but this is an emergent technology, and the bottles weren't released during our time frame in the Bay of Bengal so we cannot use these as a validation dataset in this study. The drifters referenced in the methods were undrogued and the sentence in question has been updated to state this. An explanation that we do not expect the drifters to behave as all plastics would, because plastic behaviour is affected by shape and density, has been added to the drifter validation subsection of the methods to clarify this (lines 204-210):

"Undrogued drifters would float at the surface of the ocean and are therefore analogous to some types of floating marine litter. The movement of floating litter at the surface of the ocean differs due to factors such as shape and density, particularly with respect to the effect of wind (Pereiro et al., 2018). Therefore, drifters are not expected to represent all items of floating litter, but they are one of the closest analogies that can be tracked to validate the particle tracks in the model."

Please mention the temporal resolutions of the ocean models in the 2$^{nd}$ paragraph of the Methods as well as the horizontal resolution. This is briefly mentioned later in the Methods, along with a vague reference to sensitivity tests in Appendix A. I think the temporal resolution is important, and the choice to use daily-mean (or is it daily intervals?) velocity fields rather than hourly velocity fields is non-trivial. This should be clarified and discussed up front. Regarding the sensitivity tests in Appendix A, while I do not think that these are critical to the manuscript and its results, I am afraid that I do not find the tests themselves very convincing. Looking at the particle positions in CMEMS, these are quite different between the daily and hourly resolutions (though the ROMS positions are remarkably similar, and interestingly the CMEMS daily positions seem to more closely resemble those from ROMS). Since the sensitivity simulations were only run for a month, I don't think using the beaching connectivity matrices is a valid justification for using the daily resolution. Perhaps a comparison between hourly and daily particle trajectories (as was done with the validation against drifter trajectories) makes more sense here? It is also not clear to me why the authors prefer to use daily velocity fields when hourly ones are available? I think there should be some justification for this, and it should also be mentioned in the Discussion as a possible further limitation to capturing coastal dynamics relevant to beaching.

The sentence specifying that daily-mean ocean, Stokes drift, and wind velocities forcing was used for the simulations has been moved to the second paragraph as you suggested. The drifters that were used in our study don't match the time period used for this sensitivity analysis, therefore, to include a new comparison would be a significant amount of extra effort that we are unable to do at this time. Additionally, the drifter comparison with particle movements that is detailed in the main text shows there is a considerable spread of particles released at the drifter locations due to diffusion. Given that the focus of the analysis for the manuscript was on country-to-country connectivity and beaching, as quantified by connectivity matrices, this sensitivity analysis was undertaken to ensure that daily forcing would give results consistent with hourly forcing. That is why the sensitivity study focussed on whether there was a large difference between connectivity when using daily versus hourly forcing. We appreciate that reviewer does not think this analysis is critical to results but we want to leave it in because the daily versus hourly approach has been raised in other studies. We have chosen to put this analysis in the Appendix for the reason that it is not crucial to the results we present in the main paper.

Please also mention the horizontal and temporal resolutions of the Stokes drift and wind fields in the 2$^{nd}$ paragraph in the Methods.

Thank you for pointing out this omission. The details have been added to the end of this paragraph (lines 128-132):

"Stokes drift velocities and wind fields at a height of 10 m above land, respectively. Stokes drift velocities were available in 3-hourly timesteps at a resolution of 1/5°, which is roughly 21 km at the latitudes of the Bay of Bengal; wind velocities were hourly and with a resolution of 1/4°, which is approximately 26 km at these latitudes. Both datasets were interpolated onto the relevant grid for each of the CMEMS and ROMS runs using cubic interpolation and then averaged to daily timesteps."

Please note that, in addition to Chassignet et al. (2021), van der Mheen et al. (2020b) also identified country-to-country connectivity. It may be worth adding a comparison to those results as well (currently only a comparison with particles exiting the region is done, not country

connectivity). Alternatively, I would briefly mention that van der Mheen et al. (2020b) also includes this country-to-country connectivity and explain why no comparison with these results is done.

van der Mheen et al. (2020a) did publish connectivity matrices detailing country-country connections but they populated their matrices as a percentages of the total sink particles on a given country, whereas our analysis used percentages relating to the total released from a given source country, so the results cannot be compared. Given our chosen method of sourcing particles uniformly, the total number of particles released from a country is relative to the length of its coastline rather than based on any estimates of pollution. Therefore, calculating our connectivity matrices from the point of view of total sinked particles per country would not be informative.

We have added this detail to the manuscript and clarified why we cannot compare our results with theirs (lines 317-319):

"van der Mheen et al., (2020a) calculated connections between countries but chose to publish their results as the percentage of total particles that beached on a given country rather than total particles released from a given country. Therefore, our results are not comparable with their findings."

In the Discussion (L380-383) and Conclusion (L402-405) the possibility of weighting particle releases based on future improved source estimates is suggested. I personally think it unlikely that this would be done, it seems more likely that simulations would be rerun with more particles released from relevant sources, especially since running particle tracking simulations is relatively accessible and not very computationally costly. I suspect the authors mention this possibility to justify the uniform release (there is quite a lot of emphasis on justifying this throughout the manuscript), but I think this release is already well-justified because it allows a focus on the dynamics of floating plastic waste in the Bay of Bengal without having to deal with uncertainty surrounding source-estimates.

We are glad you see the benefits of the study as is, even without the need to further apply it to future research questions. Another reviewer had similar reservations about using our results in future when combined with updated estimates of plastics making their way into the ocean. Therefore, we removed the text you referred to and the Discussion and Conclusions no longer mention using the model as a tool that could be reused with newer, more accurate weightings applied to our results.

In the Abstract (L23-26) and Conclusion (L404-406), I personally think that the results of this study are generalised a bit too strongly. I would say that the main results of this study are that most simulated floating plastic beaches in its country of origin and that the EICC seems the main driver of plastic transport in the Bay of Bengal. Perhaps some more general conclusions can be drawn from this, for example that countries preventing plastic waste entering the ocean will benefit from reduced plastic waste washing up on their own shores, but I would not say that this study can be used directly to target beach clean-ups and aid policy decisions. I would suggest a bit more nuance in both the Conclusion and the final sentences of the Abstract.

Following this and comments from another reviewer, this paragraph has been removed from the conclusions section. The statement in the abstract about aiding policy decisions has also been altered to place less emphasis on how the results might be helpful to policymakers (lines 24-25):

"This study will therefore crucially inform future research and policy in this region, providing advice on the accuracy of different modelling approaches independent of assumptions of the source locations or volumes."

**Figure 1**: Can you add the months to the "Pre-monsoon & monsoon" and "Post-monsoon" labels in the figure? Is it possible to also add a general direction for the wind/Stokes drift to the schematics here?

The months for each season have been included in the season labels and arrows indicating the average direction of the wind and Stokes velocities for the periods depicted have been added to each sub figure.

**Figure 2**: Panel a shows "a snapshot of ocean speeds from the ROMS dataset". Please add the date used for the snapshot.

This information has been added to the figure caption.

Can you make the coloured dots in the legend of panel a larger? Please also check if the colours chosen are suitable for different colour vision deficiencies.

The colours of all figures have been altered to be more colour vision deficient-friendly. Thank you for the reminder! We have added extra dots to the legend so they are still the same size as those in the drifter tracks but are easier to identify.

Perhaps consider adding a "start" and "end" marker to each of the drifter trajectories (of the overall trajectories only, not the weekly portions for validation).

A large star marker has been added to indicate the starting position for each of the drifters. We feel this is sufficient to indicate the direction of the trajectories.

I appreciate the drifter trajectory with the particles in panel b. Are the particles for the CMEMS simulation? Would it be possible to show the same for the other drifters as well (perhaps in an Appendix/as supplemental figure)? D4 may be very chaotic to show, but I would be interested to at least see D1 (lowest MCSD for the ROMS run) as well as D5 (lowest MCSD for the CMEMS run).

The particles shown in Figure 2b are from the ROMS simulation, The caption has been updated to specify this. We have added a figure to the appendix which shows the validation particles for all drifters in both runs and added a note into the caption for Figure 2 to direct readers to this new figure.

**Table 2**: Please add the months to the monsoon seasons as well. Also, perhaps consider "Full simulation" instead of "Year"?

We have taken your suggestions and altered the figure as such.

Should this be Table 1?

Yes, thank you for pointing this error out. The caption has been altered.

**Figure 3**: Please also add the months to the monsoon seasons above the connectivity matrices here as well. Consider "Full simulation" instead of "Year" for the top panels.

These changes have been made.

**References**

[revised manuscript text omitted]

---

## Author Response (AR2)

**Response to reviewer's comments on "Monsoonal influence on floating marine litter pathways in the Bay of Bengal" by Harrison et al.**

We thank the editor and reviewers for their further consideration of our manuscript and have made additional changes based on the comments on our revised manuscript. Our response to each point made by the reviewer is presented below in blue.

Reviewer #1 Comments

The manuscript has certainly improved in the revision process. The corrections and comments have been duly taken into account by the authors in the revised version of the manuscript. As a result, several methodological aspects that were not entirely clear in the original submission are now easier to follow. I still have a few minor comments that I feel need to be addressed before publication:

In the abstract :

L13. I suggest removing the reference to "novel approach", I don't think this is really novel, on the contrary it goes back to a basic principle of uniform and equidistant particle release justified by a lack of information on source estimates.

We have removed the word "novel."

L22. I would also suggest changing "Both simulations were validated" to "Both simulations were evaluated": the word validation does not seem appropriate given that only a few drifter trajectories were used for this phase.

This change has been made as you suggest.

L24-25. "This study will [...] volumes" : This sentence seems unclear. Does this study really provide information on the accuracy of the modeling approaches, given that there is no validation of the observed plastic distribution to say that one simulation or hypothesis is really better than others?

This is a valid point and we have altered the wording of this sentence to make sure the usefulness of the study is clearer, without referring to accuracy:

"This study will therefore crucially inform future research and policy in this region, providing advice on the benefits and suitability of selecting different modelling approaches independent of assumptions of the source locations or volumes."

L104-108. "Windage is implemented in the model by applying 1% of the wind velocity to the particles' trajectories. Following analysis of observations of the wind's effect on undrogued drifters by Pereiro et al. (2018), this should describe all but very buoyant items of litter." : Again, I understand that the 1% windage is important for this class of very buoyant items: but what percentage of the mass of plastic debris and particles at sea does this debris represent? My impression is that the percentage of debris with a significant emerged part remains small and that the majority of floating debris drifts to the subsurface due to processes of biofouling and degradation. I therefore suggest specifying in the abstract that the study focuses only on a class of highly buoyant debris with a relatively large size, which would represent about xx% of plastic debris at sea, if the literature provides information on these proportions.

This sentence stated that the choice of 1% windage applies to all **but** the most buoyant items of litter. The most buoyant items of litter are the only class of plastic items for which we expect a windage coefficient of 1% is **not** suitable, which as you note, is likely a small percentage of floating plastic litter. It is estimated that around 65% of plastics produced are lighter than seawater and would therefore float at the surface of the ocean (Pattiaratchi et al., 2022). Our approach of using a windage coefficient of 1% is consistent with other particle tracking studies simulating floating marine litter in this region (e.g. Irfan et al., 2024; van der Mheen et al., 2019).

To avoid misunderstandings about the types of buoyant plastic we are representing in our simulations, we have exchanged the word "but" for "except" in this sentence. This should make it clearer to readers that the particles in our model represent the majority of positively buoyant plastics entering the ocean:

"...this should describe all except very buoyant items of litter."

Fig. C1: (e) and (f): the particle clouds are identical between the two simulations, which does not correspond to the difference shown in the histogram bars in Fig. 2 (c) drifter D3.

Thank you for pointing this out, the CMEMS particle clouds had accidentally been added to both subplots. The ROMS particle clouds have now been included in Fig. C1 (f) and the differences between (e) and (f) now match up with Fig. 2 (c).

L145. "weightings could be applied as a post-process in the future": seems to be a little excessive given the limitations mentioned and the low numerical cost of running Lagrangian simulations and should be removed (as previously suggested by both reviewers).

We have previously removed all other references to this application of our results from the Abstract and Conclusions sections, based on previous reviews, to scale down the emphasis placed on the future importance of our results. However, we feel dropping this point altogether reduces the value of the manuscript. While computational costs to run these experiments are indeed relatively low these days, setting up and parameterising the scientific code is a time consuming and specialised task. We are providing an option to non-specialists (i.e. model users rather than just modellers) to scale our results in future to quickly and easily compute weighted source-to-sink estimates, making them available to wider audiences and avoiding the need for modelling expertise. Therefore, we have opted to retain this sentence in the manuscript.

L275. "though" → "through"

Thank you! This typo has been fixed.

References

Irfan, T., Isobe, A., and Matsuura, H.: A particle tracking model approach to determine the dispersal of riverine plastic debris released into the Indian Ocean, Marine Pollution Bulletin, 199, 115985, https://doi.org/10.1016/j.marpolbul.2023.115985, 2024.

van der Mheen, M., Pattiaratchi, C., and van Sebille, E.: Role of Indian Ocean Dynamics on Accumulation of Buoyant Debris, Journal of Geophysical Research: Oceans, 124, 2571–2590, https://doi.org/10.1029/2018JC014806, 2019.

Pattiaratchi, C., van der Mheen, M., Schlundt, C., Narayanaswamy, B. E., Sura, A., Hajbane, S., White, R., Kumar, N., Fernandes, M., and Wijeratne, S.: Plastics in the Indian Ocean – sources,

transport, distribution, and impacts, Ocean Science, 18, 1–28, https://doi.org/10.5194/os-18-1-2022, 2022.